# Dzip1 and Fam92 form a ciliary transition zone complex with cell type specific roles in *Drosophila*

Jean-André Lapart[1], Marco Gottardo[2], Elisabeth Cortier[1], Jean-Luc Duteyrat[1], Céline Augière[1‡], Alain Mangé[3], Julie Jerber[1§], Jérôme Solassol[3], Jay Gopalakrishnan[2], Joëlle Thomas[1†*], Bénédicte Durand[1†*]

[1]Institut NeuroMyoGène, CNRS UMR 5310, INSERM U1217, Université Claude Bernard Lyon 1, Lyon, France; [2]Institute of Human Genetics, Universitätsklinikum Düsseldorf, Heinrich-Heine-Universität Düsseldorf, Düsseldorf, Germany; [3]IRCM, INSERM, Université de Montpellier, ICM, Montpellier, France

*For correspondence:
joelle.thomas@univ-lyon1.fr (JT);
durand-b@univ-lyon1.fr (BD)

[†]These authors contributed equally to this work

Present address: [‡]Centre de recherche du CHU de Québec (CHUL), Québec, Canada; [§]Open Targets, Wellcome Genome Campus, Cambridge, United Kingdom

Competing interests: The authors declare that no competing interests exist.

**Abstract** Cilia and flagella are conserved eukaryotic organelles essential for cellular signaling and motility. Cilia dysfunctions cause life-threatening ciliopathies, many of which are due to defects in the transition zone (TZ), a complex structure of the ciliary base. Therefore, understanding TZ assembly, which relies on ordered interactions of multiprotein modules, is of critical importance. Here, we show that *Drosophila* Dzip1 and Fam92 form a functional module which constrains the conserved core TZ protein, Cep290, to the ciliary base. We identify cell type specific roles of this functional module in two different tissues. While it is required for TZ assembly in all *Drosophila* ciliated cells, it also regulates basal-body growth and docking to the plasma membrane during spermatogenesis. We therefore demonstrate a novel regulatory role for Dzip1 and Fam92 in mediating membrane/basal-body interactions and show that these interactions exhibit cell type specific functions in basal-body maturation and TZ organization.

## Introduction

Cilia and flagella are highly conserved organelles present at the surface of eukaryotic cells. They play major physiological roles in animals such as cell or fluid mobility, signaling during development and cellular homeostasis. The importance of cilia and flagella is highlighted by the discovery of human diseases, classified as ciliopathies, that are associated with defects in cilia structure and/or function (*Badano et al., 2006*; *Baker and Beales, 2009*; *Brown and Witman, 2014*). Cilia and flagella are templated from the basal body, derived from the mother centriole, from which the microtubule based axoneme is assembled.

At the base of the cilium, a specific compartment, the transition zone (TZ) plays a critical role in cilium assembly and function. Many genes responsible for cilia associated diseases such as the Meckel syndrome (MKS), Joubert syndrome or nephronophthisis (NPHP) are caused by defects in proteins of the TZ (*Czarnecki and Shah, 2012*). The TZ functions as a ciliary gate by sorting selected components in and out of the cilium, thus controlling the specific composition of the ciliary compartment (*Garcia and Reiter, 2016*; *Reiter et al., 2012*; *Gonçalves and Pelletier, 2017*). TZ assembly starts during the first steps of cilia formation, when the mother centriole associates with cytoplasmic vesicles before docking to the plasma membrane. Assembly of TZ proteins is spatiotemporally controlled and requires, in most organisms, at least three different protein modules namely MKS, NPHP and CEP290. Extensive genetic, biochemical studies and super resolution microscopy analysis helped to establish a hierarchy of these components and a structural view of the TZ architecture

**eLife digest** Many animal cells have hair-like structures called cilia on their surface, which help them to sense and interact with their surroundings. The cilia are supported by protein filaments and must assemble correctly because faulty cilia can lead to several life-threatening diseases. Problems in an area at the base of the cilia, known as the 'transition zone', account for the most severe forms of these diseases in humans.

The transition zone is responsible for selecting which proteins are allowed in and out of the cilia. The transition zone itself is made up of many proteins that work together to determine the cilia composition. But not all of these proteins are known, and it is unclear how those that are known affect cilia structure.

One protein found in transition zones of several animals, including fruit flies and mice, is called Cby. Lapart et al. set out to understand which other proteins interact with Cby in fruit flies to better understand what this protein does in the transition zone. A series of experiments showed that Cby interacts with two proteins called Dzip1 and Fam92 to regulate the assembly of transition zones. Together these three proteins constrain a core component of the transition zone, a fourth protein called Cep290, to the base of the cilia.

Fruit flies only have cilia on cells in their sensory organs and testes and, in both types of tissue, cilia could only form properly when Dzip1 and Fam92 were present. Lapart et al. also showed that, in the fruit fly testes, Dzip1 and Fam92 helped to anchor the newly forming cilia to the cell surface. This anchoring role was particularly important for the fruit flies' sperm to grow their characteristic whip-like tails, which are a specialized type of cilia that allow sperm cells to move.

Overall, the findings show how some transition zone proteins work together and that they can have different effects in different tissues. Understanding the mechanisms behind healthy cilia assembly will likely be key to tackling cilia-related diseases.

(*Williams et al., 2011*; *Garcia-Gonzalo et al., 2011*; *Sang et al., 2011*; *Chih et al., 2012*; *Gupta et al., 2015*; *Garcia and Reiter, 2016*; *Reiter et al., 2012*; *Gonçalves and Pelletier, 2017*).

Although largely conserved from worms to mammals, all TZ proteins are not conserved in all ciliated species and variations exist between model organisms. For example, the NPHP module, present in mammals, worms and protozoa, is not conserved in flies, whereas CEP290 and several members of the MKS module are conserved in most organisms (*Basiri et al., 2014*; *Pratt et al., 2016*; *Vieillard et al., 2016*). In addition to the core TZ components, several others have been identified but their precise relationships with the core known TZ components are not understood. Among these other proteins, Chibby (Cby), a conserved TZ component in vertebrates and flies, is required for cilia function both in mammals and *Drosophila* (*Burke et al., 2014*; *Voronina et al., 2009*; *Enjolras et al., 2012*; *Vieillard et al., 2016*). In vertebrates, CBY has been shown to interact with several basal body (BB) associated proteins, such as ODF2 or CEP164 (*Lee et al., 2014*; *Siller et al., 2017*; *Burke et al., 2014*; *Steere et al., 2012*; *Chang et al., 2013*) and more recently DZIP1L, DZIP1 and FAM92a or b proteins (*Wang et al., 2018*; *Li et al., 2016b*; *Breslow et al., 2017*). However, the functional integration of CBY and these interactors in TZ assembly is unclear and some of those, such as Cep164 cannot likely sustain Cby function in *Drosophila*, as Cep164 does not seem to be expressed in *Drosophila* testes (Flybase).

We show here that the unique *Drosophila* orthologs Dzip1 (CG13617) and Fam92 (CG6405) of respectively, vertebrate DZIP1 or DZIP1L and FAM92a or b, interact and cooperate with Cby in flies. We demonstrate that all three proteins form a strictly ordered functional module, and cooperate in building the TZ in the two *Drosophila* ciliated tissues, with Dzip1 acting upstream of Fam92 and Cby. While our observations establish that Dzip1 and Fam92 localization at the TZ relies on Cep290, they reveal that Dzip1 and Fam92 exert a negative regulatory feedback loop by restraining Cep290 localization to the ciliary base.

Last, our work reveals remarkable differences in the role of Dzip1 and Fam92 in regulating basal bodies (BB) docking between the two *Drosophila* ciliated tissues. Whereas, loss of Dzip1 or Fam92 does not affect basal body docking in sensory cilia, it impairs BB-membrane growth and attachment in spermatocytes. As a consequence, we observed aberrant and premature elongation of the

axoneme before completion of meiosis, highlighting a primary role of the BB-membrane associated compartment for regulating axonemal microtubule elongation in *Drosophila* spermatocytes. These aberrant elongations mostly affect the daughter centrioles, revealing functional differences of the mother and daughter centrioles in *Drosophila* spermatocytes.

## Results

### Dzip1/Fam92/Cby form a complex at the ciliary transition zone in *Drosophila*

We initially identified mouse CBY1 interactors by LAP-Tag affinity purification using IMCD3 cells (murine Inner Medullary Collecting Duct cells) stably expressing CBY1 fused to the EGFP-TEV-S peptide tag (see Materials and methods section). This EGFP-TEV-S tag in N-terminus allows to detect the protein in cells and its purification by a two-step affinity procedure (*Torres et al., 2009*). Analysis of LAP-CBY1 complex revealed the presence of DZIP1 and FAM92 among 22 identified proteins (see full list in *Supplementary file 1*), in agreement with recently published data (*Li et al., 2016b*; *Breslow et al., 2018*; *Wang et al., 2018*). *DZIP1/1L* and *FAM92a/b* each show a unique ortholog gene in *Drosophila*, *CG13617* and *CG6405* respectively (*Figure 1—figure supplement 1A*), but are absent, like *CBY1*, from the *C. elegans* genome. Hereafter, we name *CG13617* and *CG6405* as *Dzip1* and *Fam92*, respectively. By co-immunoprecipitating Cby-GFP and HA-Dzip1 or HA-Fam92, we demonstrate that *Drosophila* Dzip1 and Fam92, each interact with Cby (*Figure 1—figure supplement 1B–C*). Dzip1 or Fam92 do not apparently interact with each other in these co-IP experiments (*Figure 1—figure supplement 1D*). However, when all three proteins are expressed together, immunoprecipitation of GFP-Dzip1 pulls down both Cby and Fam92, suggesting that all three proteins are present in one complex when co-expressed in mammalian cells (*Figure 1—figure supplement 1D*).

To identify the subcellular localization of *Drosophila* Dzip1 and Fam92, we generated transgenic flies expressing Dzip1-GFP or Fam92-GFP under the control of their respective promoters (*Figure 1—figure supplement 2*). We determined that both *dzip1* and *fam92* are exclusively expressed in the two kinds of *Drosophila* ciliated cell types, namely type I sensory neurons and male germ cells (*Figures 1* and *2*). Type I sensory neurons comprise chordotonal (Ch) and external sensory (ES) neurons, which harbor motile and immotile cilia respectively (*Figure 1A*) (*Gogendeau and Basto, 2010*; *Jana et al., 2016*). Each sensory neuron is enclosed in several support cells forming the sensory organ or scolopidia. Dzip1 and Fam92 decorate the base of the cilia at the tip of the sensory dendrites (labeled with 22C10) (*Figure 1B*, arrows). By performing, super-resolution 3D structured-illumination microscopy (3D-SIM), we confirmed that Dzip1 and Fam92 co-localize with Cby in sensory neurons (*Figure 1C*), demonstrating their restricted localization at the ciliary transition zone.

In the male germline, Dzip1 and Fam92 appear first in spermatocytes (*Figure 2*) at the distal end of centrioles. In spermatocytes, centrioles have a specific behavior as both mother and daughter centrioles of each pair elongate and dock to the plasma membrane. All four basal bodies (BB) extend a TZ, also described as a primary cilium-like structure (*Figure 2A*) (*Pasmans and Tates, 1971*; *Riparbelli et al., 2012*; *Gottardo et al., 2013*; *Vieillard et al., 2016*). Subsequently, during meiosis, all four basal bodies are engulfed inside the cytoplasm, together with the primary like cilium which hence creates a ciliary cap that ensheaths each basal body distal end. After meiosis and at the onset of axoneme elongation, the ciliary cap is remodeled and a distinct domain, the ring centriole, appears at its base (*Vieillard et al., 2016*; *Basiri et al., 2014*; *Fabian and Brill, 2012*). The axoneme grows inside the ciliary cap, which concomitantly migrates, extruding the nascent axoneme out in the cytoplasm (*Figure 2A*) (*Pasmans and Tates, 1971*; *Riparbelli et al., 2013*; *Gottardo et al., 2013*). We observed that Dzip1, Fam92 and Cby all localize at the centriolar distal ends in early spermatocytes (*Figure 2B–C*). During centriolar/BB maturation, as the ciliary cap grows, Dzip1 and Fam92 localizations are extended and overlap with Cby, as revealed by 3D-SIM observations (*Figure 2D*). In spermatids, when the TZ migrates away from the basal body, we observed that Dzip1 and Fam92 strongly accumulate with Cby at the ring centriole (*Figure 2B–C*, arrows).

Together, these results strongly indicate that Dzip1/Fam92/Cby interact at the ciliary transition zone.

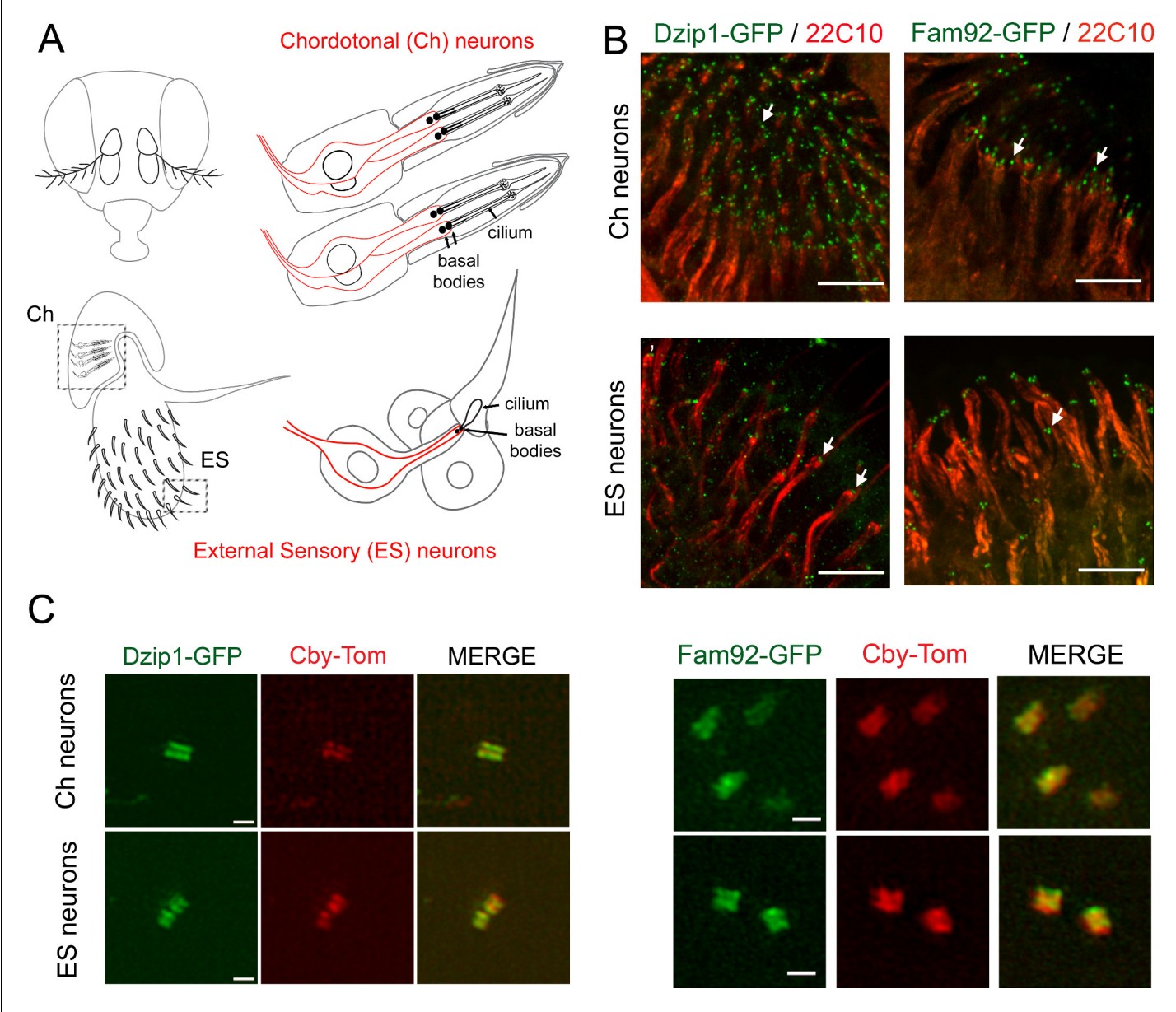

**Figure 1.** *Drosophila* Dzip1 and Fam92 are expressed in ciliated sensory neurons and locate at the ciliary transition zone. (A) Scheme of the two types of sensory organs (or scolopidia) present in the antennae of the *Drosophila* head. Chordotonal scolopidia are proprioceptors present in the second antennal segment of the *Drosophila* head and respond to sound vibrations and gravity. Each chordotonal scolopidia is composed of several support cells (gray) and comprise two or three ciliated neurons (red). External sensory organs (or scolopidia) are present under each sensory sensilla of the third antennal segment and react to olfactory and chemical stimuli. In other body parts, ES organs also respond to mechanical stimuli. External sensory scolopidia comprise support cells (gray) and only one ciliated neuron (red). In ciliated sensory neurons, the two centrioles (or proximal and distal basal bodies) stand above each other at the tip of the dendrites. (B) Whole-mount staining of the second and third antennal segment showing neuronal cell bodies and dendrites (22C10), Dzip1-GFP and Fam92-GFP. Dzip1-GFP and Fam92-GFP are present at the tip of the dendrites in each type of neurons. Arrows point both to Dzip1-GFP and Fam92-GFP localization at the tip of the dendrites. (C) 3D-SIM imaging of Ch and ES neurons. Both *Drosophila* Dzip1 and Fam92 overlap with Cby at the transition zone. Bars = 10 μm for (B); = 0.5 μm for (C).

The online version of this article includes the following figure supplement(s) for figure 1:

**Figure supplement 1.** *Drosophila* Dzip1 and Fam92 interact with Cby.

**Figure supplement 2.** *dzip1* and *fam92* loci and genetic tools.

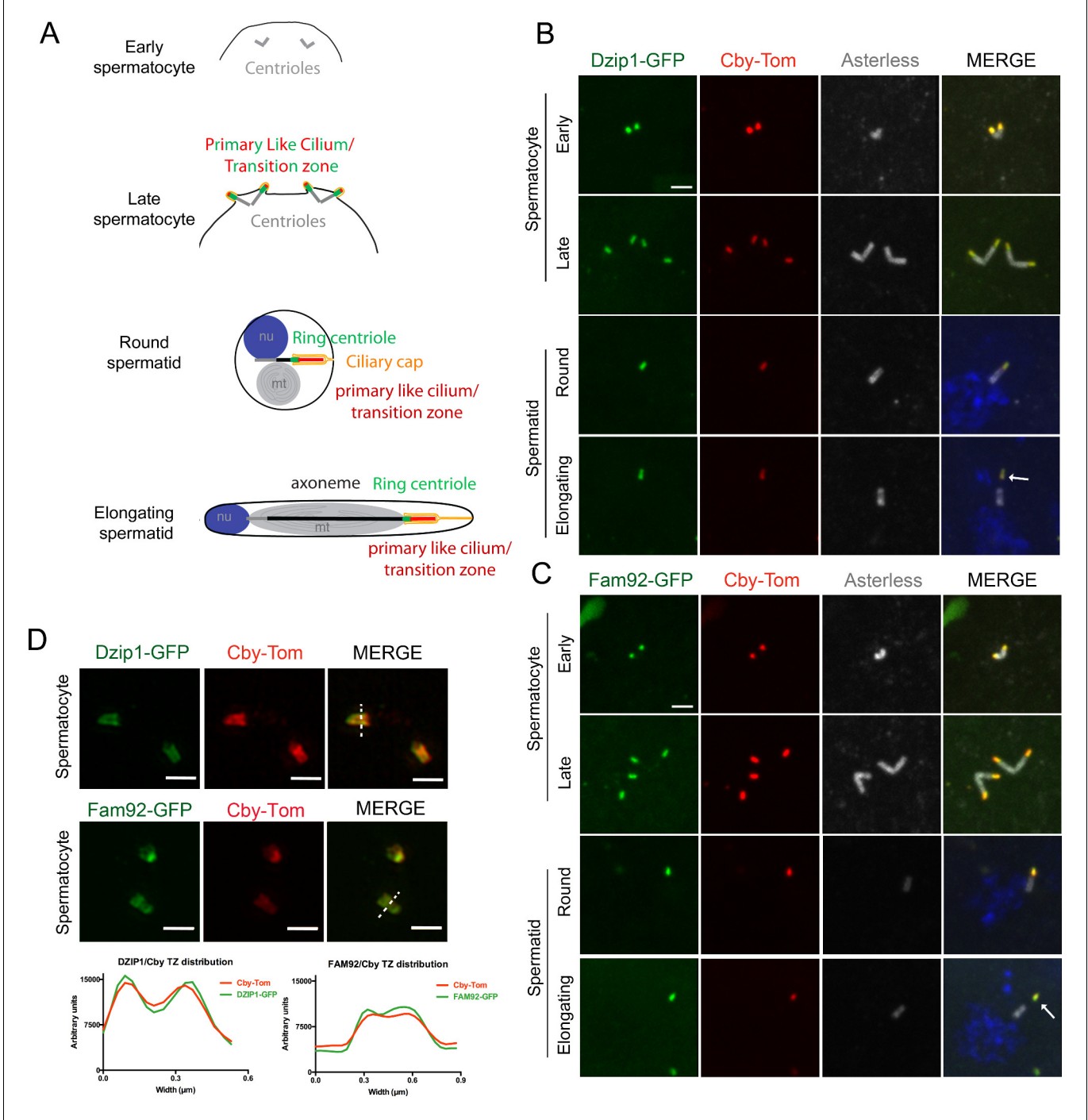

**Figure 2.** *Drosophila* Dzip1 and Fam92 are located at the transition zone during spermatogenesis. (**A**) Scheme illustrating centriole/basal bodies (BB) behavior during spermatogenesis in male germ cells. In early spermatocytes, two pairs of centrioles (gray) are present in each cell. During spermatocyte maturation, centrioles convert to BB and dock to the plasma membrane while extending a primary like cilium which is entirely decorated with TZ proteins (green and red). In late spermatocytes, the TZ, also described as the primary like cilium, reaches approximatively 500 nm and protrudes at the cell surface. During meiosis, BB with primary like cilia/TZ are internalized, thus retaining the ciliary membrane (yellow) connected to the plasma membrane, hence creating a membrane cap in spermatids. In round spermatid, BB are apposed to the nuclear membrane and the ciliary cap (yellow) connected to the plasma membrane is extended. The ring centriole (green) marks the base of the ciliary cap which is decorated by TZ proteins. At the onset of axoneme elongation, axonemal microtubule nucleation inside the cap extends the ciliary cap/TZ, and as the axoneme grows (black), the ring centriole is pushed away from the basal body. (**B–C**) Confocal imaging of whole-mount testes showing Dzip1-GFP (**B**), Fam92-GFP (**C**), Cby-Tom and Asterless (centrioles). Dzip1 and Fam92 appear together with Cby in early spermatocytes at the tip of centrioles. In elongating spermatids, *Drosophila*

*Figure 2 continued on next page*

*Figure 2 continued*

Dzip1 and Fam92 mark the ring centriole (arrows) separating from the BB (asterless, gray). (D) 3D-SIM imaging of male germ cells. *Drosophila* Dzip1 and Fam92 overlap with Cby in spermatocytes. Plots of the intensity profile of the centrioles along the dotted lines illustrate the overlay between Cby-Tom and Dzip-GFP or Fam92-GFP. Bars = 1 μm.

## Dzip1 and Fam92 are required for cilia and flagella formation in *Drosophila*

To determine the roles of the Dzip1/Fam92 module in TZ assembly or function, we generated two deletion alleles of *dzip1* or *fam92* (*Figure 1—figure supplement 2* and Materials and method section). In the *dzip1¹* allele, the 65 N-ter codons (including start codon) were deleted and replaced by a 3 kb insertion including the mini-white gene (*Figure 1—figure supplement 2*). In *fam92¹*, exon two and part of exon three were removed (294 bp, *Figure 1—figure supplement 2*), leading to the deletion of 81 aa and a frameshift with several stop codons in the remaining downstream sequence, leaving only 47 aa of the WT protein and 39 amino acids of the −1 frameshifted open reading frame.

*dzip1¹* and *fam92¹* flies are viable but show typical behavioral defects associated with ciliary dysfunction in *Drosophila*. Fly geotaxis response (*Figure 3A*) was monitored by bang assay (*Enjolras et al., 2012*). Whereas most of the control flies reach and stay at the top of the tube 30 s after the bang, no *dzip1¹* flies reached the top of the tube (*Video 1*). This phenotype was worsened in *dzip1¹/Df* as the flies not only fail to climb, but are completely immotile with held up wings (*Video 2*), indicating that *dzip1¹* is likely not a complete null allele. Both *dzip1¹* and *dzip1¹/Df* behavioral phenotypes were fully rescued by one copy of *dzip1::GFP* (not shown). In contrast, a few percentage (17%) of *fam92¹* flies could reach the top of the tube (*Figure 3A*) and no differences could be observed when comparing *fam92¹* flies and *fam92¹* over its cognate deficiency (*fam92¹/Df*), indicating that *fam92¹* is likely a null allele. The *fam92::GFP* rescue construct partially restores the climbing behavior of the *fam92¹* mutant flies. These observations reveal a graded requirement for Fam92 and Dzip1 in *Drosophila* geotaxis response and indicate a regulatory role of this complex in sensory cilia assembly or function.

To determine the impact of Dzip1 or Fam92 loss on cilia assembly, we performed ultra-structural analysis by transmission electron microscopy (EM). Strikingly, we observed that cilia were essentially absent in *dzip1¹* antennal chordotonal organs, but were still present in *fam92¹* antennae (*Figure 3B and F*), in agreement with their milder geotaxis defect behavior. In *fam92¹* flies, 25% of scolopidia (total observed n = 54) showed reduced cilia number (one or no cilia). In addition, defects in axonemal structure were occasionally observed, such as lack of microtubule doublets (*Figure 3C*, asterisks, 6%), excess accumulation of material beneath the ciliary membrane (arrows) and deformation of the membrane (*Figure 3C*, upper panel, dot, 4%). Most TZ sections showed normal ultrastructure, but a few (6%) showed incomplete radial symmetry and accumulating material as observed for cilia. Linkers connecting the axoneme to the membrane could still be observed (*Figure 3C*, arrowhead, lower panel). Transition zones were completely disorganized in *dzip1¹* antennae as revealed by serial-cross and longitudinal sections (*Figure 3D–E*) and no more axoneme-to-membrane linkers could be observed. Basal bodies were normally present at the dendrite distal tip (*Figure 3D*, lower panel), but we observed a rapid disorganization of the axoneme, with its complete abrogation a few microns above the basal body (*Figure 3D–E*).

Hence, these observations demonstrate that loss of Dzip1 or Fam92 affects TZ assembly required for sensory cilia formation.

To address the function of Dzip1 and Fam92 in sperm flagellum assembly, we first investigated male fertility. We observed a strong reduction of the fertility of *fam92¹* males compared to controls. This defect is restored by expressing Fam92-GFP (*Figure 4A*). Because *dzip1¹* mutant flies are severely uncoordinated, their fertility could not be tested. However, *dzip1¹* testes showed a marked dispersion of the nuclei along the cysts and, as a consequence, altered migration of sperm individualization complexes (*Figure 4B* and *Figure 4—figure supplement 1A*). Nuclear dispersion is also observed, to a lesser extent, in *fam92¹* testes (*Figure 4B* and *Figure 4—figure supplement 1A*). One possible origin of nuclei dispersion is defective axonemal elongation (*Vieillard et al., 2016*; *Soulavie et al., 2014*) and EM analysis confirmed that axonemes were affected in *fam92¹* cells

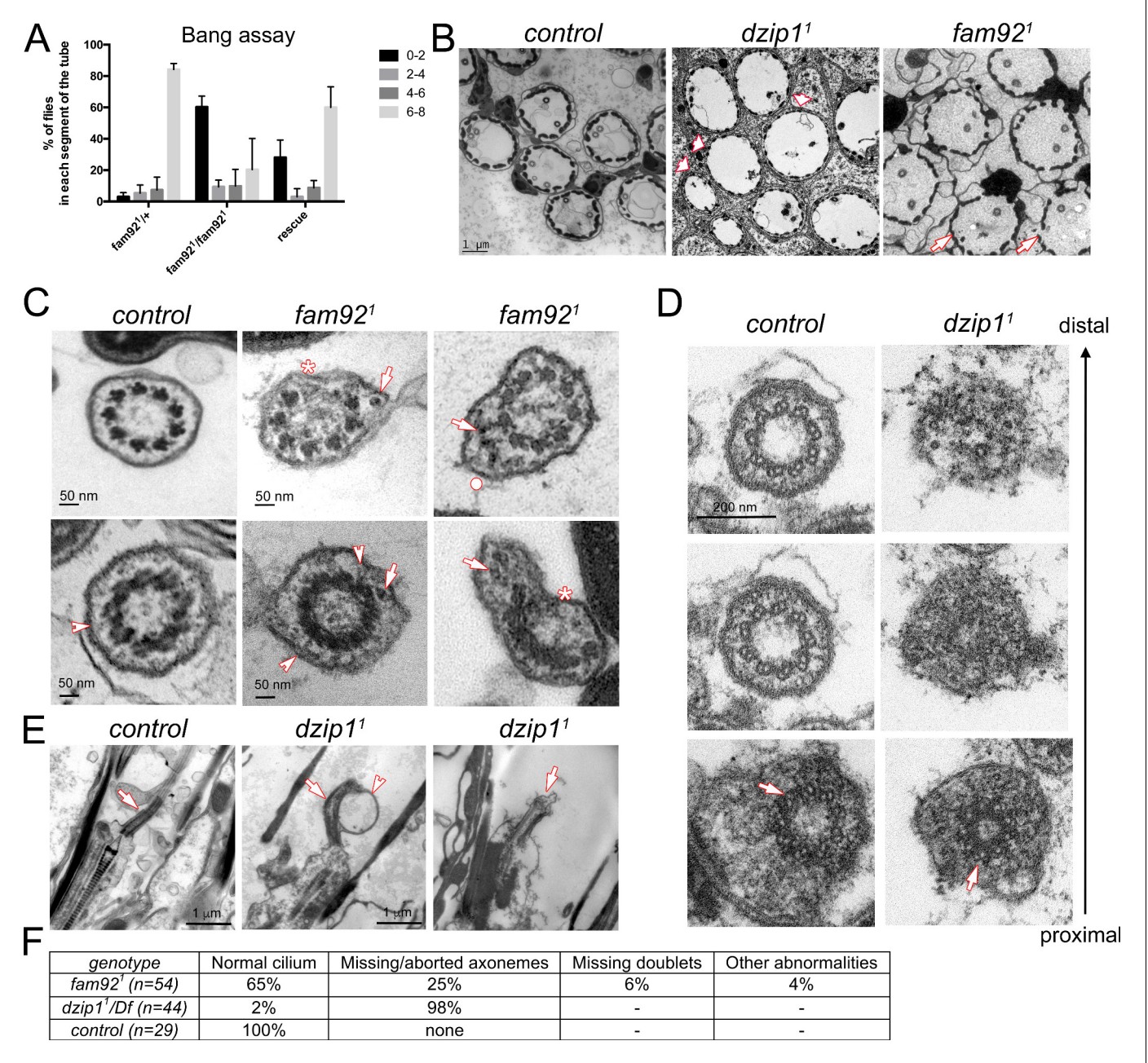

**Figure 3.** Dzip1 and Fam92 are required for ciliogenesis in *Drosophila sensory cilia*. (**A**) Quantification by bang assay of geotaxis response of *fam92[1]* flies compared to control or rescued flies. The percentage of flies that reach a defined level in the tube (0–2, 2–4, 4–6, 6–8) is represented. *fam92[1]* flies are unable to climb along the tube compared to control flies. This defect is partially rescued by adding two copies of *fam92::GFP* transgene (*fam92[1]/+* n = 61, *fam92[1]/fam92[1]* n = 63, *rescue* n = 68). (**B**) EM analysis of cross sections of antennal Chordotonal (Ch) neurons. Whereas two neurons/cilia can be observed in each control scolopidia, cilia are almost completely absent in *dzip1[1]* scolopidia (arrowheads). In *fam92[1]*, reduced number of cilia are observed on several Ch neurons (arrows). (**C**) Ch neurons cilia ultrastructure of *fam92[1]* antennae, showing reduced number of microtubule doublets (asterisks) and/or accumulation of dense material (arrows) underneath the ciliary membrane and deformation of the membrane (dot). Similar defects are also observed on cross sections of the TZ (lower panels). Note that the linkers connecting the axoneme to the membrane are still present (arrowheads). (**D**) Serial sections of the basal body-transition zone region of Ch neurons in *dzip1[1]* compared to control, from the basal body (proximal, lower panels) to the TZ (distal, upper panels). Whereas doublet microtubules are present and symmetrically organized at the basal body level (arrows, lower panels), they fail to elongate along the transition zone which is incompletely assembled. (**E**) Longitudinal sections of *dzip1[1]* showing basal body with aberrant TZ compared to control (arrows). Membrane bulges (arrowhead) along the aberrant *dzip1[1]* TZ can also be detected. (**F**) Quantifications of the cilium defects observed in scolopidia.

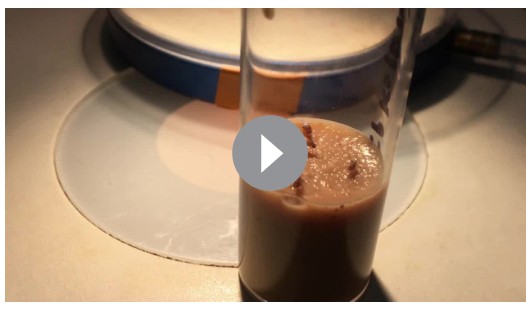

**Video 1.** Real time imaging of *dzip1*[1] adult flies.
https://elifesciences.org/articles/49307#video1

(*Figure 4D*, 125/448 broken axonemes, 25/448 missing central pair, 52/448 missing axoneme). As well, in *dzip1*[1] or *dzip1*[1]*/Df* mutant testes, marked axonemal defects could be observed. In *dzip1*[1] testes, we observed 6/126 broken axonemes, 1/126 missing central pair, 1/126 missing axonemes (*Figure 4C*). More severe defects were observed in very young *dzip1*[1]*/Df* spermatid cysts, with up to 60% of spermatids with missing or broken axonemes (*Figure 4—figure supplement 1B–C*). However, in older cysts, the number of spermatids/cyst was much reduced compared to control (mean = 44 spermatids/cyst compared to control = 63.4), but almost all remaining elongated spermatids showed normal axonemes (*Figure 4—figure supplement 1B*). These observations suggest that young spermatids with severe axonemal defects fail to elongate and are not observed in cross sections of older spermatid cysts.

Taken together, these results demonstrate that Fam92 and Dzip1 are necessary at the ciliary TZ for cilia formation both in sensory neurons and male germ cells.

## Dzip1 and Fam92 form a strictly ordered functional module with Cby

To determine the functional hierarchy between Dzip1, Fam92 and Cby, we analyzed their respective localization in *dzip1*[1] or *fam92*[1] or previously described *cby*[1] mutants (*Figure 5* and *Figure 5—figure supplement 1*).

Cby-Tom is almost completely lost from ciliary TZ in *dzip1*[1]*/Df* sensory neurons (*Figure 5—figure supplement 1A*). In spermatocytes, Cby-Tom signal is strongly reduced at the tip of centrioles (*Figure 5A,D*) but, in rare occasions, expanded at one centriole of the pair (asterisk, 2%). The amount of Fam92-GFP is severely reduced in *dzip1*[1]*/Df* sensory cilia base (*Figure 5—figure supplement 1B*) and in spermatocytes (*Figure 5B,D*, arrows). Altogether, these observations indicate that Dzip1 is required at the basal body/TZ to recruit or stabilize both Cby and Fam92.

In *fam92*[1] mutant flies, Dzip1-GFP is slightly reduced at the distal tip of basal bodies in sensory cilia (*Figure 5—figure supplement 1C*). In *fam92*[1] spermatocytes, Dzip1-GFP domain is sometimes expanded (8.3% of cases, *Figure 5D*), but the overall Dzip1-GFP expression is also slightly reduced (*Figure 5C*, arrows, 5D). Cby-Tom is completely lost in both *fam92*[1] ciliated tissues (*Figure 5A*, arrowheads and *Figure 5—figure supplement 1A*). Hence, Fam92 is required to recruit Cby, but not Dzip1, at the TZ. As well, in *cby*[1] tissues, Fam92-GFP completely disappears from basal bodies (*Figure 5B*, arrowhead and *Figure 5—figure supplement 1B*), indicating that Cby and Fam92 stabilize each other at the basal body. In *cby*[1] mutant spermatocytes, an extended domain of Dzip1-GFP is occasionally observed (*Figure 5C*, arrows, 4.5%, 5D). However, no overall difference in Dzip1-GFP intensity was observed in *cby*[1] mutant testes (*Figure 5C–D*) or antennae (not shown) compared to controls.

These observations establish a functional hierarchy for Dzip1, Fam92 and Cby: Dzip1 is required to recruit or stabilize both Fam92 and Cby at centrioles, but does not depend on Fam92 or Cby for its targeting to centrioles, the latter only restricting Dzip1 to the proximal TZ. Conversely, Fam92 and Cby mutually depend on each other to localize at the TZ.

## Dzip1 and Fam92 restrict Cep290 to the proximal TZ

To understand how Dzip1 and Fam92 organize the ciliary base, we investigated their functional relationships with other core TZ components.

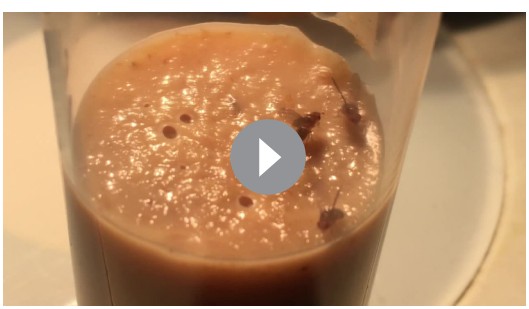

**Video 2.** Real time imaging of *dzip1*[1]*/Df* adult flies.
https://elifesciences.org/articles/49307#video2

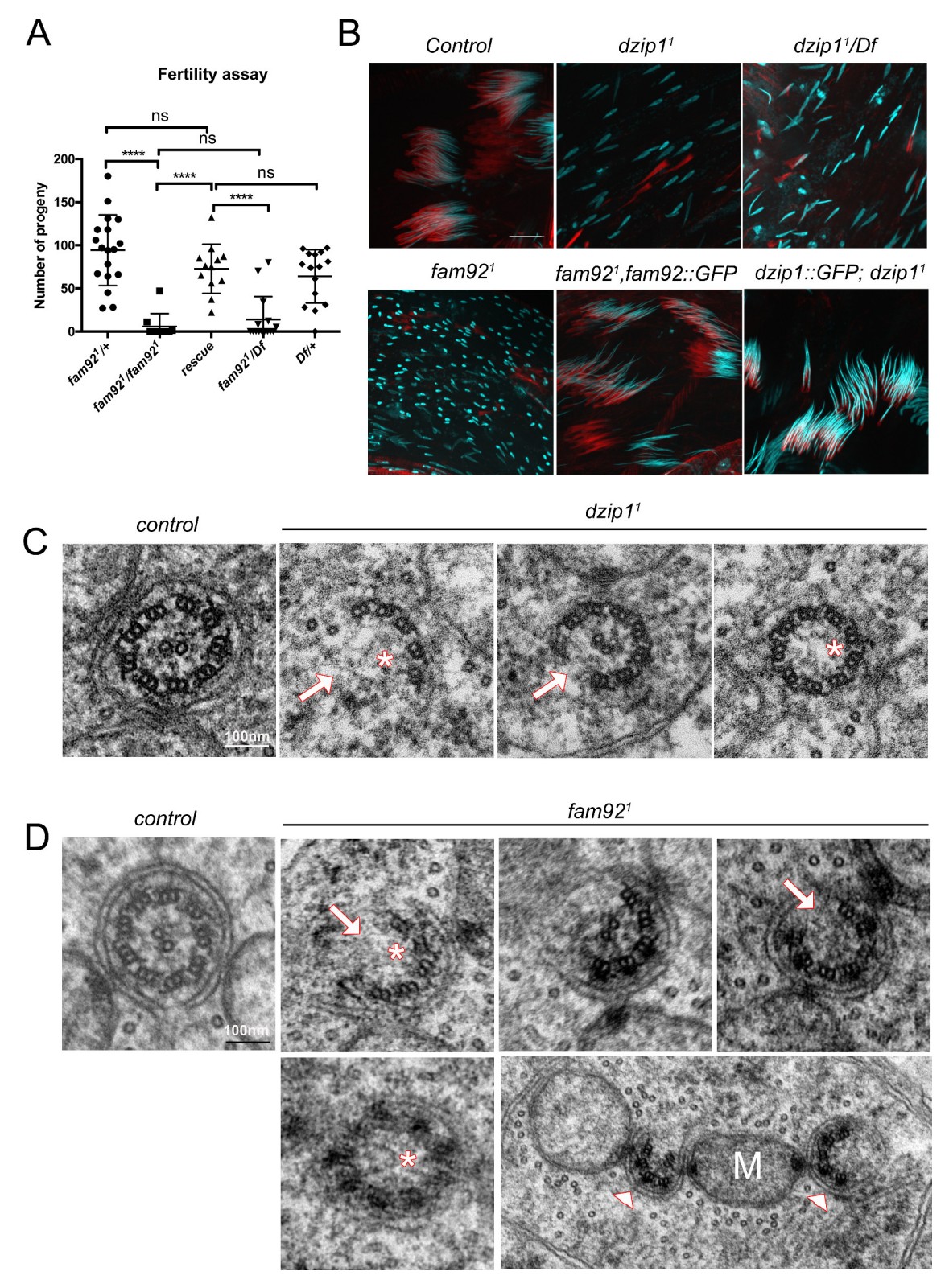

**Figure 4.** Dzip1 and Fam92 are required for flagella formation. (**A**) Fertility assays of *fam92¹* males showing severe reduction of fertility in *fam92¹* or *fam92¹/Df* compared to control or rescued flies (*fam92¹/+* n = 18; *fam92¹/fam92¹* n = 10; rescue n = 15; *fam92¹/Df* n = 15; *Df/+* n = 16) \*\*\*\*, p<0.0001; ns, p>0,05. (**B**) Confocal analysis of whole mount testes of control, *dzip1¹*, *dzip1¹/Df*, *fam92¹* and rescued genotypes stained for nucleus (Hoechst, cyan) and actin cones (phalloidin, red). Nuclei are dispersed in mutant cysts compared to controls, which could reflect axonemal elongation defects. As a

*Figure 4 continued on next page*

*Figure 4 continued*

consequence, actin cones required for spermatid individualization are mislocalized. Axoneme elongation and spermatid individualization are normal in rescued flies. Bars = 10 μm. (C) EM analysis of cross sections of axoneme of round spermatid. In *dzip1¹*, flagella ultrastructure is altered with missing microtubule doublets (arrows) or central pair (asterisks). (D) In *fam92¹* mutant spermatids, most axonemes show altered ultrastructure, with missing microtubule central pairs (asterisks) or doublets (arrows), broken symmetry with each part of the axoneme being relocated along the mitochondria (M, arrowheads).

The online version of this article includes the following figure supplement(s) for figure 4:

**Figure supplement 1.** *Dzip1¹* and *fam92¹* testis phenotypes.

We first investigated the behavior of Mks1, a component of the core conserved MKS complex (*Weatherbee et al., 2009*). In *Drosophila*, Mks1 is required to assemble the MKS complex, but removal of MKS components leads to only very mild defects of cilia assembly (*Vieillard et al., 2016*; *Pratt et al., 2016*). In *dzip1¹/Df* and *fam92¹* mutant flies, Mks1-GFP is lost or reduced at the TZ in both ciliated tissues (*Figure 6A*, arrows, and *Figure 6—figure supplement 1A*). Dzip1 and Fam92 are hence important for the recruitment or stabilization of the MKS module at the TZ.

Cep290 is a core conserved TZ component which plays a prominent role in TZ assembly in many organisms, including *Drosophila* (*Craige et al., 2010*; *Rachel et al., 2015*; *Li et al., 2016a*; *Basiri et al., 2014*). In flies, Cep290 is located at the base of the TZ in sensory neurons and is a critical component of the migrating ring centriole during spermatogenesis (*Basiri et al., 2014*). In the absence of Dzip1, besides a small but significant reduction of Cep290-GFP on spermatocyte centrioles, we observed striking Cep290-GFP expanded domains, both in spermatocytes and in Ch neurons (*Figure 6A–B*, arrows). Around 13% of centrioles in spermatocytes and a majority in antennae showed an expanded Cep290-GFP domain (*Figure 6A–B*). In *fam92¹* mutants, we also observed a few Cep290-GFP expanded domains in spermatocytes and antennae, but with no significant difference in overall Cep290-GFP intensity (*Figure 6A–B*, arrows).

To further understand the relationships between Cep290, Dzip1 and Fam92, we took advantage of a strong *cep290* hypomorphic mutant, *cep290⁰¹⁵³⁻ᴳ⁴*. This mutant shows severe uncoordination and completely disorganized spermatid cysts with dispersed nuclei (*Figure 6—figure supplement 2A*), which is a consequence of the severe axonemal elongation defects observed in this mutant (not shown). The phenotypes were completely rescued by two copies of *cep290::GFP* (*Figure 6—figure supplement 2A–B*). We observed that in *cep290⁰¹⁵³⁻ᴳ⁴* flies, Dzip1 and Fam92 are strongly reduced or lost at the TZ, both in spermatocytes (*Figure 6C*) and in sensory neurons (*Figure 6—figure supplement 1B*).

These observations demonstrate that Cep290 is required during TZ assembly to recruit Dzip1 and Fam92, which in turn restrict Cep290 scaffolding to the proximal part of the TZ.

## Dzip1 and Fam92 facilitate basal body docking to the plasma membrane in spermatocytes

Defects in BB anchoring to the plasma membrane in spermatocytes lead to aberrant growth of axonemal microtubules (*Vieillard et al., 2016*). We used the specific axonemal marker CG6652-GFP that only labels axonemal microtubules in flies (*Figure 6—figure supplement 2C–D*). With this marker, we observed aberrant growth of axonemal microtubules in spermatocytes, with graded severities increasing from *fam92¹*, *dzip1¹* to *cep290⁰¹⁵³⁻ᴳ⁴* mutant flies (*Figure 7A–B*). 30% of centrioles showed CG6652 extensions in *fam92¹* testes, 34% in *dzip1¹/Df* and 76% in *cep290⁰¹⁵³⁻ᴳ⁴*. More strikingly, whereas 68% of the centriole pairs present extensions from both centrioles in *cep290⁰¹⁵³⁻ᴳ⁴* spermatocytes, we observed that in *fam92¹* testes, 96% of the centriole pairs present microtubule extensions on only one centriole. This asymmetric penetrance of the phenotype is related to centriole age. Indeed, among 24 centriole pairs for which mother and daughter identities could be unequivocally assigned, 19 daughter centrioles (79%) and only five mother centrioles (21%) were affected. As well, in *dzip1¹/Df* testes, among 40 centriole pairs, we found 30 daughter centrioles (75%) and 10 mother centrioles (25%) affected by the absence of Dzip1 (*Figure 7A–B*).

We analyzed the ultrastructure of the centrioles and primary like cilium in the initial stages of elongation (*Figure 7C*). We observed irregular distal end (asterisk) of centrioles in *dzip1¹* testes at young spermatocytes stages before docking (upper panel). Whereas centrioles dock in polar

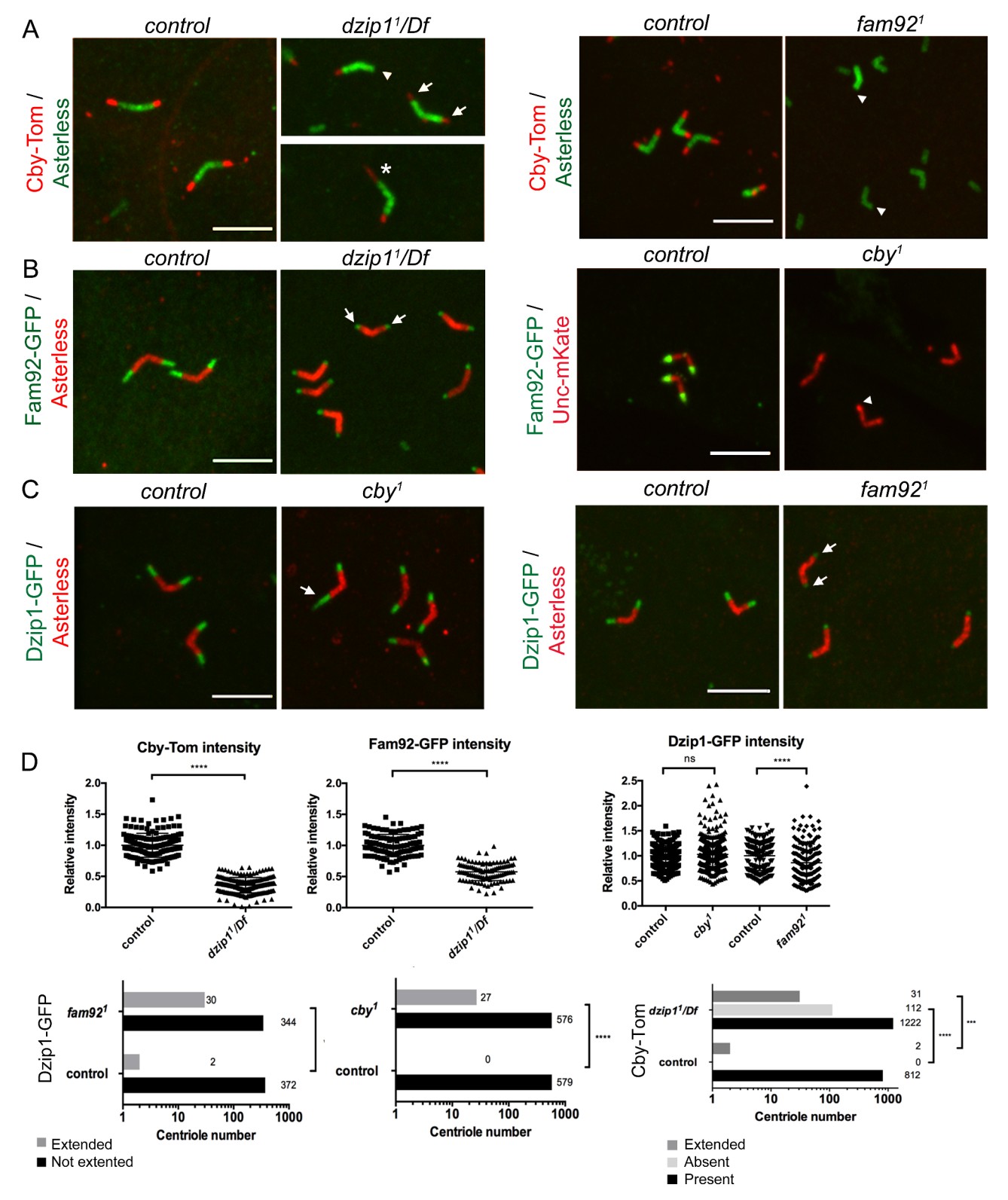

**Figure 5.** Dzip1, Fam92 and Cby work coordinately at the TZ. (**A**) Cby-Tom is present at the tip of the centrioles in control spermatocytes but strongly reduced in *dzip1¹/Df* (arrows) and can even be lost at one centriole of the pair (arrowhead), but in a few situations (2.3% see D quantifications below), Cby-Tom domain is extended (asterisk). In *fam92¹* mutant, Cby-Tom is completely lost at the tip of the spermatocyte centrioles (arrowheads). (**B**) Fam92-GFP is reduced at both centrioles of the pair in *dzip1¹/Df* (arrows) compared to control and completely lost at the tip of the centrioles

*Figure 5 continued on next page*

*Figure 5 continued*

(arrowhead) in *cby[1]* mutants. (**C**) Dzip1-GFP is strikingly expanded in a few *cby[1]* spermatocyte centrioles (arrow, 4.4% see quantifications in D) and is reduced at the tip of both centrioles in *fam92[1]* testes (arrows). Bars = 5 μm. (**D**) Quantifications of the distribution of Dzip1-GFP, Fam92-GFP and Cby-Tom in the different mutant contexts. ****, p<0.0001; ***, p<0.001; ns, p>0,05.

The online version of this article includes the following figure supplement(s) for figure 5:

**Figure supplement 1.** Dzip1, Fam92 and Cby are cooperatively recruited at the TZ in *Drosophila* sensory cilia.

spermatocytes before reaching their full size in control testes, undocked centrioles were observed in *dzip1[1]*. Either both centrioles of the pair were undocked or partially docked (n = 2, lower panel) (*Figure 7C*, arrowhead) or only one centriole of the pair, the mother, was docked (n = 3) (*Figure 7C*, arrow). These results show that Dzip1 is required to cap the centriole distal end and foster its anchoring to the plasma membrane. As well, we observed undocked centrioles in *fam92[1]* mutant spermatocytes, with irregular distal end (asterisk) or with microtubules extending from the distal end (*Figure 7C*, arrow), illustrating the role of Fam92 in controlling centriolar distal growth and docking to the plasma membrane.

Altogether, these observations demonstrate that Dzip1 and Fam92 are required for the proper distal elongation of basal bodies and their membrane anchoring in *Drosophila* spermatocytes. This docking is required to regulate TZ elongation (*Figure 7D*). In addition, our results show that in male germ cells, although all centrioles have the capacity to generate a ciliary cap, a functional asymmetry of the mother and the daughter centrioles is revealed by their ability to dock to the plasma membrane in absence of key TZ proteins.

## Discussion

Our work establishes the functional hierarchy of Dzip1, Fam92 and Cby, which thus define an intrinsic transition zone module for the initiation of ciliogenesis in *Drosophila*. Dzip1 is required to recruit Fam92 and Cby at the basal body distal end, allowing proper TZ assembly. Dzip1 and Fam92 act downstream of the core TZ protein Cep290 but also regulate its accumulation at the distal basal body. Our work sheds light on tissue specific variations in the initiation of ciliogenesis in the two *Drosophila* ciliated tissues. In testes, basal body docking strictly depends on the integrity of Dzip1 and Fam92. However, in sensory neurons alterations of the latter complex affect only TZ assembly and cilia formation but not BB docking. Last, our work demonstrates that Dzip1 and Fam92 control the distal elongation of basal bodies and their docking to the plasma membrane, which thus regulate the onset and proper elongation of axonemal microtubules in *Drosophila* male germ cells.

### Conservation of TZ assembly pathways in eukaryotes

In vertebrates, two orthologs have been described for Dzip1 and Fam92 and three for Cby. DZIP1, DZIP1L, FAM92a and b also interact with CBY1 in vertebrates (*Wang et al., 2018*; *Breslow et al., 2017*; *Ye et al., 2014*) indicating a conservation of the interacting capacities of the family members during evolution. The precise hierarchy between members of the complex has not been established in mammals, but depletion of CBY1 prevents the recruitment of FAM92a/b to the centriole (*Li et al., 2016b*) and DZIP1L was shown to act upstream of CBY1 (*Wang et al., 2018*; *Keller et al., 2009*).

In vertebrates, mutations in members of this module lead to ciliogenesis defects of various severities and with different phenotypic outcomes (*Wolff et al., 2004*; *Tay et al., 2010*; *Glazer et al., 2010*; *Li et al., 2016b*; *Breslow et al., 2017*) *Dzip1* or *Dzip1L* KO mice die during embryogenesis (*Wang et al., 2018*; *Wang et al., 2013*), whereas *Fam92a[-/-]* mice show skeletal defects (*Schrauwen et al., 2019*) and *Cby1* KO mice exhibit differentiation defects of motile ciliated epithelia (*Burke et al., 2014*; *Voronina et al., 2009*). This phenotypic variability is not expected if all three proteins only act together in a functional module at the TZ, as demonstrated by our work in *Drosophila*. This suggests that the different mouse paralogs of Cby and Fam92 may have acquired specialized ciliogenic functions in mouse. However, we also observed that *dzip1[1]* and *fam92[1]* mutant phenotypes show small differences indicating specific functions of each proteins. For instance, in both ciliated tissues, *dzip1[1]* hypomorphic mutant phenotype is more severe than *fam92[1]*, suggesting that Dzip1 has additional functions that are not solely mediated by Fam92.

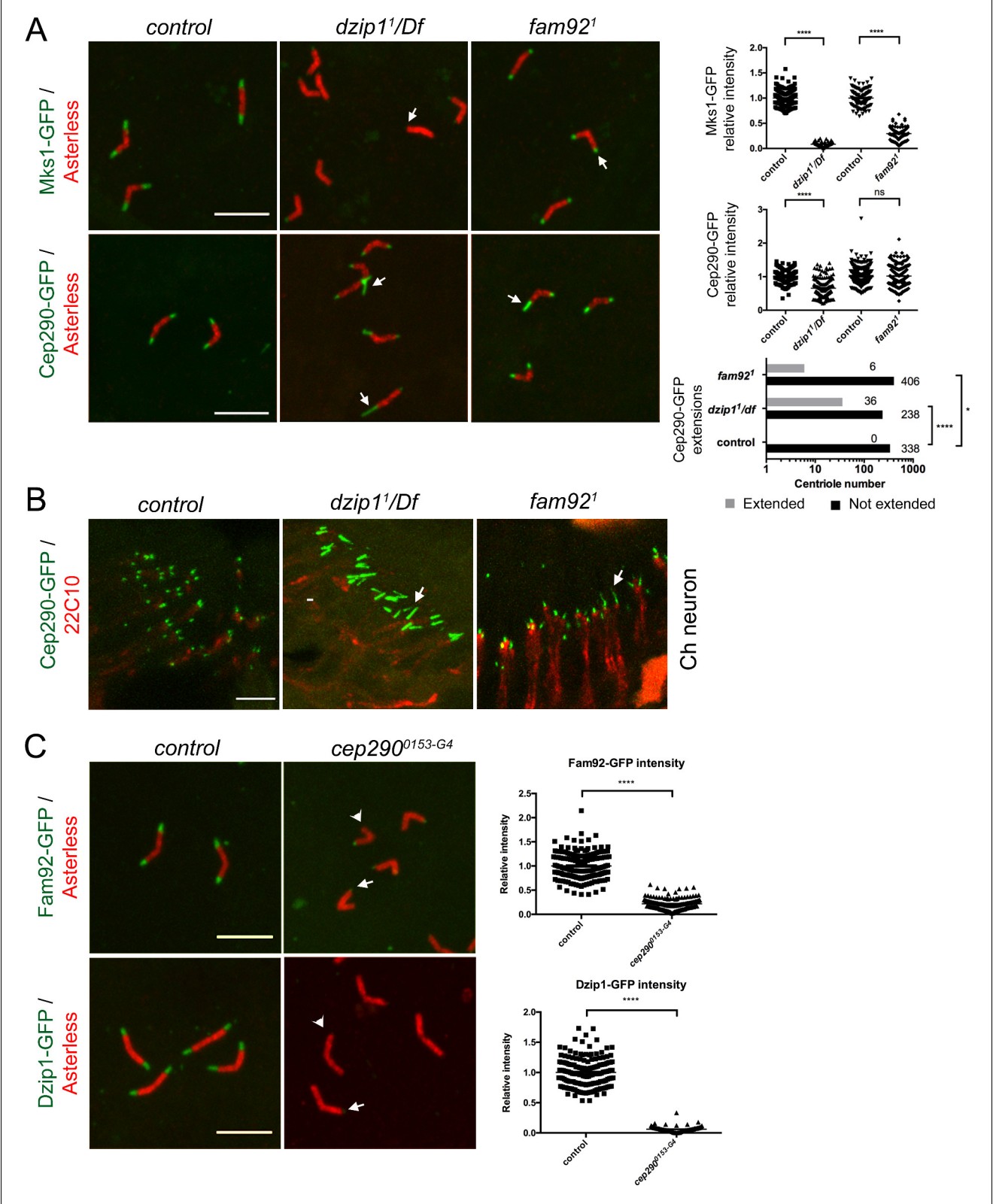

**Figure 6.** Dzip1 and Fam92 organize the transition zone downstream of Cep290. (**A**) In spermatocytes, Mks1-GFP is severely reduced at the ciliary cap or lost in *dzip1[1]/Df* and *fam92[1]* (arrows, quantifications on right graph). Cep290-GFP domain is expanded in *dzip1[1]/Df* (arrows), but less affected in *fam92[1]* (quantifications on right graph). (**B**) Cep290-GFP is expanded in chordotonal cilia of *dzip1[1]/Df* or *fam92[1]* antennae (arrows). (**C**) Fam92-GFP and

*Figure 6 continued on next page*

*Figure 6 continued*

Dzip1-GFP are strongly reduced (arrows) or lost (arrowheads) at the tip of the centrioles in *cep290^{0153-G4}* spermatocytes (quantifications are illustrated on the graphs). ****, p<0.0001; *, p<0.01; ns, p>0.05. Bars = 5 µm.

The online version of this article includes the following figure supplement(s) for figure 6:

**Figure supplement 1.** Dzip1 and Fam92 organize the TZ downstream of Cep290 in sensory cilia.

**Figure supplement 2.** *cep290^{0153-G4}* is a strong hypomorphic allele and CG6652 labels the axoneme.

Despite the conserved role of Dzip1, Cby and Fam92 in TZ and cilia assembly from *Drosophila* to humans, no homologs could be detected in the genomes of *C. elegans*. This could be linked to the diversification of cilia function, with both motile and immotile cilia being present from *Drosophila* to humans, in contrast to *C. elegans*, where only one subtype of cilia that are immotile is found. Another possible explanation could be that DZIP1/FAM92/CBY are associated with specific signaling or developmental functions in animals that still need to be understood.

## Tissue specific functional properties of the TZ proteins in BB docking

This work emphasizes the essential role of Dzip1 and Fam92 in building the ciliary transition zone in the two types of ciliated tissues of *Drosophila*. Strikingly, it also reveals tissue specific function of these proteins in priming basal body/membrane docking in *Drosophila* testes. This reveals intrinsic differences in the mechanisms that link basal body to membranes in *Drosophila* ciliated tissues. In mammals, basal body docking requires transition fibers built from the distal appendages on the mother centriole prior to docking (*Wei et al., 2015*). In *Drosophila*, distal appendages have not been observed on centrioles, but structures similar to transition fibers are described at the base of sensory cilia (*Ma and Jarman, 2011*; *Vieillard et al., 2016*), whereas only scarce links could be observed in male germ cells (*Gottardo et al., 2018*; *Roque et al., 2018*). These differences could explain why destabilization of the TZ leads to basal body anchoring defects in spermatocytes, but not in sensory neurons. This structural characteristic of the spermatocyte TZ is likely to be related to its specific functional properties. Indeed, whereas the TZ is stably built at the ciliary base in sensory neurons, it shows a dynamic behavior during sperm flagella extension, separating from the basal body and migrating along the growing end of the axoneme (*Basiri et al., 2014*; *Avidor-Reiss et al., 2017*; *Baker et al., 2004*; *Wei et al., 2008*). In addition, in spermatocytes, basal bodies have a dynamic behavior, being first docked at the plasma membrane and next internalized during meiosis (*Pasmans and Tates, 1971*; *Fabian and Brill, 2012*). This could induce mechanical constraints on basal bodies that would increase their sensitivity to TZ disruption. In agreement with this hypothesis, when TZ maturation was challenged by modulating membrane phospholipids (*Gupta et al., 2018*), BB were released from the plasma membrane during meiosis, but their initial docking was not impaired. However, we observed by EM, that BB fail to initially dock in significant occurrences (8 among 13) in *dzip1^1* and *fam92^1* mutant spermatocytes, indicating that Dzip1 and Fam92 are at least involved in the initial steps of BB docking. Previous observations of another strong hypomorphic *cep290^{mecH}* allele showed docked basal bodies in spermatocytes and spermatids (*Basiri et al., 2014*). In *cep290^{0153-G4}* mutant, we did not quantify the number of docked versus undocked basal bodies in spermatocytes, but we observed up to 76% of aberrant axonemal growth, suggesting that basal body to membrane attachment is compromised in this mutant.

Differences in the organization of the ciliary base associated with variations in the distribution and function of several centriolar and TZ proteins have been documented in *Drosophila* ciliated cells (*Jana et al., 2018*). However, none of these identified differences help to explain the behavioral properties of BB docking and TZ dynamics that we have identified in the two ciliated *Drosophila* tissues. Hence, additional screens for specific factors of basal-body docking or TZ assembly either in sensory neurons or male germ cells need to be designed.

## Mother and daughter centriole asymmetry in ciliogenesis

In all our observations, there is a striking phenotypic difference between the mother and daughter centrioles in spermatocytes. In all mutants examined (i.e. *dzip1*, *fam92* and *cby*) we observed a more penetrant defect on the daughter centriole than the mother. Thus, although the 2 centrioles of each pair are able to form cilia, the daughter centriole appears more sensitive to transition zone

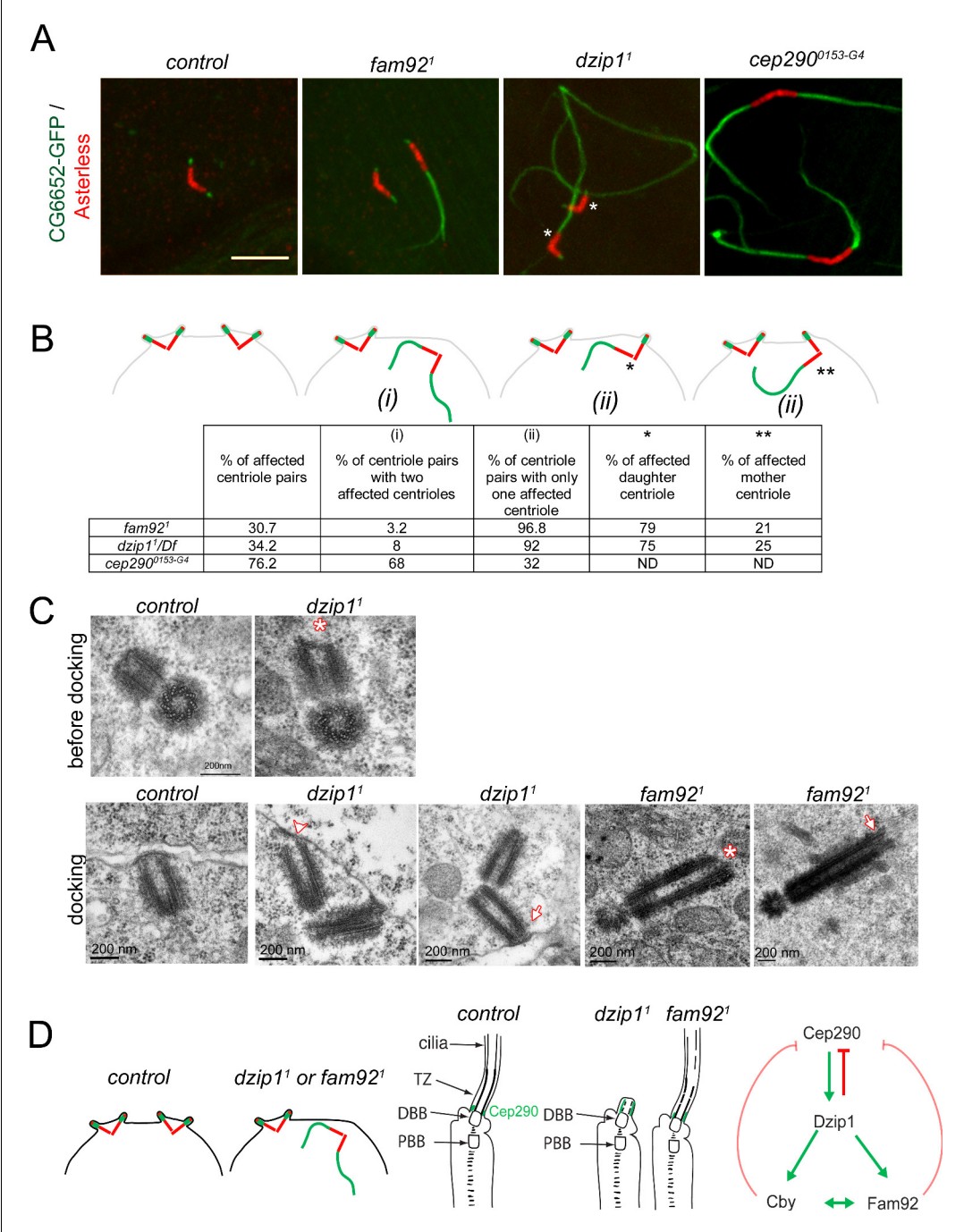

**Figure 7.** Dzip1 and Fam92 are required for both centriolar docking and proper axonemal formation during *Drosophila* spermatogenesis. (**A**) Confocal imaging of whole-mount testes showing aberrant extensions of axonemes labeled by CG6652-GFP in *fam92[1]*, *dzip1[1]* and *cep290[0153-G4]* spermatocytes. Asterisks point to the centriole in each pair that can be unambiguously assigned as the daughter. Bar = 5 μm (**B**) Quantifications of the penetrance of CG6652-GFP labeled aberrant axonemes. Schemes representing the different classes that were quantified in the table: (i) two centrioles of the pair are affected; (ii) only one centriole of the pair is affected, either the daughter* or the mother**. (**C**) EM analysis of cross and longitudinal sections of centrioles in spermatocytes. In *dzip1[1]*, centriole docking is impaired with partial docking of the centriole (arrowhead) or docking to the plasma membrane of only one centriole of the pair (arrow). Centrioles present an altered/irregular cap, compared with control (asterisk). In *fam92[1]*, undocked centrioles are also observed in spermatocytes and show irregular distal end (asterisk) or microtubules extending from the tip (arrow). (**D**) Scheme of the consequences of *dzip1* and *fam92* loss of function on basal body and cilium assembly in *Drosophila* ciliated tissues. Whereas basal body anchoring and TZ assembly is affected in male germ cells, only TZ and cilium assembly is affected in sensory cilia. In addition, Cep290-GFP is systematically expanded

*Figure 7 continued on next page*

*Figure 7 continued*

in *dzip1* mutant and occasionally expanded in *fam92* mutant chordotonal neurons. The scheme on the right summarizes the functional interactions observed between Dzip1, Cby, Fam92 and Cep290. Red arrows are inhibitory interactions and green arrows define positive interactions.

perturbations. We do not have molecular explanations for these intrinsic differences of mother and daughter centrioles in spermatocytes. In *Drosophila* sensory neurons, centrobin plays a critical role in maintaining the daughter centriole fate, precluding its capacity to build a cilium (*Gottardo et al., 2015*). In sperm cells, centrobin is required for the formation of the C tubule (*Reina et al., 2018*), which also plays a critical role in TZ assembly (*Gottardo et al., 2018*). However, in spermatocytes centrobin is equally distributed at the base of both the mother and daughter centrioles and does not show a functional asymmetry. Recently, a transient microtubule based structure that anchors the base of the mother centriole on one of the centriole pair at the onset of meiosis was identified, but its function is unknown (*Riparbelli et al., 2018*). This structure could stabilize the mother centriole and favor its attachment to the membrane, but the function of this microtubule rootlet in *Drosophila* spermatocytes needs further investigations. The intrinsic difference of the mother versus daughter centrioles could also be related to the timing of centriole docking during spermatogenesis (*Gottardo et al., 2018*), where the mother centriole was shown to dock first. This timing difference would be sufficient to better stabilize the TZ of the mother centriole, and hence explain the phenotypic differences observed in our study. There are other situations in the animal kingdom were both mother and daughter centrioles build a cilium. Among the best studied are the bi-flagellated *Chlamydomonas* and the peculiar case of multiple ciliated epithelia, where numerous de novo centrioles are assembled just at the onset of ciliogenesis. Interestingly, in mammals, CBY1 was shown to play a critical role in multiple ciliated cells to allow proper docking of the multiple basal bodies to the plasma membrane (*Burke et al., 2014*). It is tempting to speculate that the increased susceptibility of multiple ciliated cells to CBY1 loss, is related to a particular status of these de novo centrioles, as we observed for daughter centrioles in male germ cells. More work will be needed to understand the molecular determinants of the mother versus daughter centriolar functional asymmetry in *Drosophila* male germ cells.

## Specific regulatory role of the ciliary cap in *Drosophila* male germ cells

We observed that defects of TZ assembly and/or basal body docking lead to aberrant elongation of axonemal microtubules as revealed by the specific *Drosophila* axonemal marker CG6652-GFP. These abnormal elongation defects appear only in late G2 phase, just at the onset of meiosis, indicating that specific signals, yet to be identified, enable centrioles to start axonemal elongation at the onset of meiosis. The membrane cap restricts this capacity by inhibiting axonemal growth until the round spermatid stage, where a second signal turns on axonemal elongation and TZ migration. Among the candidate proteins recruited by the membrane cap which may coordinate axonemal assembly are microtubule depolymerizing kinesins as previously proposed (*Vieillard et al., 2016*). This membrane cap could also be involved in stabilizing centriolar capping proteins such as CP110 or CEP97, which in mammals need to be removed from centrioles to allow ciliary assembly (*Spektor et al., 2007*). However, there are no clear evidence in *Drosophila* that these proteins need to or are specifically removed before axonemal elongation (*Galletta et al., 2016*; *Franz et al., 2013*; *Delgehyr et al., 2012*). Our observations indicate that regulation of axonemal assembly and cell cycle regulation are tightly linked in these dividing cells, but the effectors of this control still need to be identified.

In conclusion, our work demonstrates the critical role of the conserved Dzip1/Fam92/Cby module downstream of Cep290 in initiating the assembly of the ciliary transition zone in flies. It also reveals key tissue specific differences in basal body docking pathways in *Drosophila*.

## Materials and methods

**Key resources table**

*Continued on next page*

*Continued*

| Reagent type (species) or resource | Designation | Source or reference | Identifiers | Additional information |
|---|---|---|---|---|
| Reagent type (species) or resource | Designation | Source or reference | Identifiers | Additional information |
| Gene (*Drosophila melanogaster*) | *cby* | **Enjolras et al., 2012** | FLYB: FBgn0067317 | |
| Gene (*Drosophila melanogaster*) | *fam92/CG6405* | This study | FLYB: FBgn0032428 | |
| Gene (*Drosophila melanogaster*) | *dzip1/CG13617* | This study | FLYB: FBgn0039201 | |
| Strain, strain background (*Escherichia coli*) | DH5alpha | Thermo Fisher Scientific | 18265017 | |
| Genetic reagent (*D. melanogaster*) | *Cep290^{0153-G4}* mutant strain | Bloomington Drosophila Stock Center | BDSC: 62671; FLYB: FBst0062671 RRID:BDSC_62671 | FlyBase symbol: w1118; PBac{IT.GAL4} cep2900153-G4 |
| Genetic reagent (*D. melanogaster*) | Df(3R)Exel8178 | Bloomington Drosophila Stock Center | BDSC : 7993 FBab0038335 RRID:BDSC_7993 | |
| Genetic reagent (*D. melanogaster*) | Df(2L)Exel6033 | Bloomington Drosophila Stock Center | BDSC : 7516 FLYB : FBab0037871 RRID:BDSC_7516 | |
| Genetic reagent (*D. melanogaster*) | *sas4^{s2214}* mutant strain | **Basto et al., 2006** | BDSC : 12119 FLYB : FBal0196943 RRID:BDSC_12119 | |
| Genetic reagent (*D. melanogaster*) | *cby^1* mutant strain | **Enjolras et al., 2012** | FLYB: FBal0270281 | |
| Genetic reagent (*D. melanogaster*) | *dzip1^1* mutant strain | This study | | Section Materials and methods "Generation of *dzip1^1* and *fam92^1* alleles" |
| Genetic reagent (*D. melanogaster*) | *fam92^1* mutant strain | This study | | Section Materials and methods "Generation of *dzip1^1* and *fam92^1* alleles" |
| Genetic reagent (*D. melanogaster*) | *Unc::mkate* | **Vieillard et al., 2016** | FLYB : FBal0324713 | |
| Genetic reagent (*D. melanogaster*) | *Cby::Tom* | **Vieillard et al., 2016** | FLYB : FBal0270280 | |
| Genetic reagent (*D. melanogaster*) | *Cep290::GFP* | **Basiri et al., 2014** | FLYB : FBal0301636 | |
| Genetic reagent (*D. melanogaster*) | *CG6652::GFP* | **Vieillard et al., 2016** | FlyB: FBal0324714 | |
| Genetic reagent (*D. melanogaster*) | *mks1::GFP* | **Vieillard et al., 2016** | FLYB : FBal0324710 | |
| Genetic reagent (*D. melanogaster*) | *fam92::GFP* | This study | | Section Materials and methods "Plasmids and *Drosophila* reporter gene constructs" |
| Genetic reagent (*D. melanogaster*) | *dzip1::GFP* | This study | | Section Materials and methods "Plasmids and *Drosophila* reporter gene constructs" |
| Cell line (*Mus musculus*) | IMCD3 | ATCC | CRL-2123 | |
| Cell line (*Homo-sapiens*) | HEK293 | ATCC | CRL-1573 | |

*Continued*

| Reagent type (species) or resource | Designation | Source or reference | Identifiers | Additional information |
|---|---|---|---|---|
| Cell line (*Cercopithecus aethiops*) | COS-7 | ATCC | CRL-1651 | |
| Antibody | Mouse monoclonal anti-Futsch/22C10 | DHSB | AB_528403 | IF (1:250) |
| Antibody | Rabbit polyclonal anti-GFP | Abcam | AB6556 | IF(1:1000) WB (1:10000) |
| Antibody | Guinea pig polyclonal anti-Asterless | *Klebba et al., 2013* | | IF (1:50000) |
| Antibody | Rat polyclonal anti-Asterless | *McLamarrah et al., 2018* | | IF (1:50000) |
| Antibody | Rat monoclonal anti-HA (clone 3F10) | Roche | 11867423001 | WB (1:5000) |
| Antibody | Goat polyclonal anti-rabbit Alexa 488 | Invitrogen | A11008 | IF (1:1000) |
| Antibody | Goat polyclonal anti-rabbit Alexa 647 | Invitrogen | A21244 | IF (1:1000) |
| Antibody | Goat polyclonal anti-guinea pig Alexa 488 | Invitrogen | A11073 | IF (1:1000) |
| Antibody | Goat polyclonal anti-guinea pig Alexa 594 | Invitrogen | A11076 | IF (1:1000) |
| Antibody | Donkey polyclonal anti-guinea pig Alexa 647 | Invitrogen | 706-605-148 | IF (1:1000) |
| Antibody | Goat polyclonal anti-rat Alexa 488 | Invitrogen | A11006 | IF (1:1000) |
| Antibody | Goat polyclonal anti-rat Alexa 555 | Invitrogen | A21434 | IF (1:1000) |
| Antibody | Goat polyclonal anti rat Alexa 647 | Invitrogen | A21247 | IF (1:1000) |
| Antibody | Mouse monoclonal anti-V5 | Invitrogen | R960-25 | WB (1:5000) |
| Antibody | Goat polyclonal anti-rabbit-HRP | Biorad | 170–6515 | WB (1:10000) |
| Antibody | Goat polyclonal anti-mouse-HRP | Biorad | 170–6516 | WB (1:3000) |
| Antibody | Goat polyclonal anti-rat-HRP | Sigma | A5795-1ML | WB (1:20000) |
| Antibody | Mouse monoclonal anti-γ-Tubulin | Sigma | GTU88 | IF (1:500) |
| Chemical compound, drug | phalloidin FluoProbes 547 | Interchim | FP-AZ0330 | IF (1:200) |
| Chemical compound, drug | phalloidin FluoProbes 505 | Interchim | FP-AZ0130 | IF (1:200) |
| Commercial assay or kit | GFP-TRAP | Chromotek | gta-100 | |
| Commercial assay or kit | Mouse monoclonal Anti-HA-agarose antibody | Sigma | A2095 | |
| Commercial assay or kit | S-protein agarose | Merck | 69704 | |

*Continued on next page*

*Continued*

| Reagent type (species) or resource | Designation | Source or reference | Identifiers | Additional information |
|---|---|---|---|---|
| Commercial assay or kit | Gibson Assembly Master Mix | New England Biolabs, Inc | E5510S | |
| Recombinant DNA reagent | pBFv-U6.2 | *Kondo and Ueda, 2013* Nig-Fly | FLYB: FBmc0003127 | |
| Recombinant DNA reagent | pBFv-U6.2B | *Kondo and Ueda, 2013* Nig-Fly | FLYB: FBmc0003128 | |
| Recombinant DNA reagent | pG-LAP3 vector | *Torres et al., 2009* | Addgene#79704 | |
| Recombinant DNA reagent | pEGFP-N1 | Clontech | Cat #6085–1 | |

All primer sequences are described in *Supplementary file 2*.

## Plasmids and *Drosophila* reporter gene constructs

CBY1 coding sequence was PCR amplified from mouse ependymal primary cell cDNA with primers F-CBY1 and R-CBY1. PCR product was cloned into pDONR223 (Invitrogen) and then subsequently Gateway cloned into the pG-LAP3 vector (*Torres et al., 2009*; gift from P. Jackson, Addgene#79704). This vector contains the double EGFP-TEV-S peptide tag in N- terminus allowing a two step affinity purification.

Coding sequences of *Drosophila cby*, *fam92/CG6405* and *dzip1/CG13617* were obtained by PCR on cDNA from testis. *Cby* cDNA was cloned in pEGFP-N1 (Clontech) in frame with GFP (primers F-CbyGFP/R-CbyGFP) and in pCDNA3.1-HA (gift from S. Khochbin, Institut Albert Bonniot, Grenoble ; primers F-HACby/R-HACby). cDNA of *fam92/CG6405* and *dzip1/CG13617* were cloned in pCDNA3.1-HA with primers F-Fam92HA/R-Fam92HA and F-Dzip1HA/R-Dzip1HA, respectively. pCDNA3.1-GFP, GFP-Fam92 and GFP-Dzip1 were obtained by replacement of the HA tag with EGFP. V5-Fam92 was obtained by replacement of the HA tag with V5 tag.

*Drosophila* reporter gene construct of *dzip1*/CG13617 was obtained by cloning 1.6 kb upstream regulatory sequences and the entire coding sequence (primers F-Dzip1GFPrep/R-Dzip1GFPrep) in frame with GFP of pJT61, a pattB plasmid with an extra EGFP-6xMycTag-SV40polyA cassette (*Vieillard et al., 2016*). The 4 kb fragment of *fam92/CG6405* including 1.4 kb of upstream regulatory sequences, the entire coding sequence fused to the multipurpose tag cassette including GFP and the 3'UTR was obtained by PCR with primer F-Fam92GFPrep/R-Fam92GFPrep on fosmid from the FlyFos library (*Sarov et al., 2016*). The resulting constructs were integrated in the 53B2 VK00018 landing site on the second chromosome by PhiC31 integrase (BestGene). All transgenic lines were obtained from BestGene Inc.

## Generation of *dzip1*[1] and *fam92*[1] alleles

The *dzip1*[1] allele (*CG13617*) was generated by CRISPR/Cas9 induced homologous directed repair (*Gratz et al., 2015*). Two gRNA were selected using the http://crispor.tefor.net/crispor.py website: 5'- CCCGTTTCACGGACCATCTG CGG −3' and 5'-GTTTCCAGCACTGTGCCCAG TGG −3' (proto-spacer adjacent motifs are underlined). Oligos were phosphorylated by T4PNK (New England Biolabs, Inc) and annealed. Double-stranded 5' gRNA and 3' gRNA were cloned in the BbsI site of pBFv-U6.2 and pBFv-U6.2B vectors, respectively (*Kondo and Ueda, 2013*). 5' gRNA was further subcloned in the EcoRI–NotI sites of pBFv-U6.2B to express the two gRNAs from one vector.

The 5' and 3' homology arms (1.7 kb and 2.3 kb respectively) were amplified by PCR (primers F-5'armDzip1/R-5'armDzip1 and F-3'armDzip1/R-3'armDzip1) and cloned, respectively, into the pJT38 plasmid (pRK2 plasmid [*Huang et al., 2008*], with an attB cassette) using Gibson Assembly Master Mix (New England Biolabs, Inc). The two vectors (gRNAs and homology arms) were injected into *vasa::Cas9* embryos. Flies were crossed to *w; TM2, e Ubx/TM6B, e Hu Tb* virgin females and the offspring were screened for red-eyed-flies. Homologous recombination was checked by PCR.

The *fam92*[1] allele (*CG6405*) was generated by CRISPR/Cas9 induced deletion. Two gRNA were selected as before: 5'- CATAAGACCTTGCAGATATC GGG −3' and 5'- GGCTGTCATAGCGCGGGA

TA AGG −3' (protospacer adjacent motifs are underlined). Double-stranded phosphorylated 5' gRNA and 3' gRNA were cloned into the pBFv-U6.2B plasmid. The vector was integrated into the 89E11 VK00027 landing site on the third chromosome by PhiC31 integrase (BestGene). Surviving G0 males were individually crossed to $y^2$ $v^1$ virgins. A single male transformant from each cross was mated to $y^2$ $cho^2$ $v^1$; Pr Dr/TM6C, Sb Tb virgins.

Males carrying a U6-double gRNA transgene were crossed to nos-Cas9 females to obtain male founder animals. Each male founder was crossed to three virgin females $y^2$ $cho^2$ $v^1$ ; Sco/CyO. Deletion in the fam92 locus was characterized by PCR with primers F-Fam92KO/R-Fam92KO on genomic DNA from the resultant offspring. Flies showing a 294 bp deletion in the fam92 locus were selected for further studies.

## Cell culture and transfection

All reagents for cell culture were purchased from Thermo Fisher Scientific. IMCD3 cells (murine Inner Medullary Collecting Duct cells, ATCC CRL-2123 a gift from A. Benmerah, Institut Imagine, Paris) were cultured in DMEM/HAM'S F12 medium supplemented with 10% FBS, 1X non-essential amino acids, 100 U/ml penicillin-streptomycin and 100 µg/ml hygromycin. COS-7 cells (ATCC CRL-1651) or HEK293 (ATCC CRL-1573) were maintained in DMEM medium containing 10% FBS, 1X non-essential amino acids and 100 U/ml penicillin-streptomycin. Cells were tested negative for mycoplasma.

Transfections were performed using jetPRIME (Polyplus transfection) according to the manufacturer's instructions.

## Stable cell lines

IMCD3 cell lines expressing LAP-CBY1 or control LAP-GFP were created using the Flp-In system kit (Invitrogen) and established method (*Torres et al., 2009*).

## Tandem affinity purification and mass spectrometry of CBY1-interacting proteins

Two rounds of purification of CBY1-protein complexes were performed. Cells were seeded in 35 15 cm dishes and grown to confluence before serum starvation for 24 hr to induce primary cilia formation. Cells were washed once with ice-cold PBS1X and purification was performed as described previously (*Nachury, 2008*). Briefly, cells were collected by scraping and resuspended in 11 ml lysis buffer containing HEPES pH7.5 50 mM, EGTA pH8 1 mM, MgCl2 1 mM, KCl 300 mM, Glycerol 10%, NP-40 0.31% and 1X protease inhibitor cocktail (Roche). CBY1-complexes were purified using GFP-TRAP (Chromotek). S-Tag-CBY1 was cleaved off GFP beads by TEV cleavage and the eluate was further purified on S-protein agarose (Merck) and eluted in 100 µl 2X Laemmli Sample Buffer.

For protein identification, samples were separated by 10% SDS-PAGE. Gels were stained with Coomassie and cut into four slices. Each gel slice was washed, reduced with 10 mM DTT, alkylated with 55 mM iodoacetamide, and subjected to in-gel trypsin digestion overnight (Trypsin Protease, MS Grade Pierce, Thermo Fisher Scientific). The extracted tryptic peptides were cleaned up using OMIX C18 100 µl pipette tips (Agilent), and lyophilized before being reconstituted for the LC-MS/MS analysis.

The peptides were separated using an Eksigent Ultra nano-LC HPLC system coupled with an AB Sciex Triple TOF 5600 mass spectrometer. The LC separations were performed using a Discovery Bio Wide Pore HPLC column (C18, 3 µm, 100 × 5 mm). The mobile phases used were 0.1% formic acid in water (A) and 100% acetonitrile with 0.1% formic acid. The gradient elution steps were performed with a flow rate of 5 µl/min as follows: 0–40% B for 106 min, 40–80% for 5 min, and then 80% B for an additional 5 min. All data were acquired using Analyst software (AB Sciex) in the data dependent mode. Peptide profiling was performed using a mass range of 350–1600 Da, followed by a MS/MS product ion scan from 100 to 1500 Da. For each survey MS1 scan (accumulation time of 250 msec), MS/MS spectra (accumulation time of 75 msec per MS/MS) were obtained for the 30 most abundant precursor ions. The protein identification was performed with the ProteinPilot Software 5.0 (AB Sciex). The MS/MS spectra obtained were searched against the mouse UniProt Proteome database (release 2015_09 with 46470 proteins). The search parameters for tryptic cleavage and accuracy are built-in functions of the software. The other data analysis parameters were as follows: sample type: identification; Cys-alkylation: Iodoacetamide; Digestion: Trypsin; Instrument:

TripleTOF 5600; Special factor: gel based ID, and biological modifications; Species: *Homo sapiens*; Search effort: Thorough ID. Proteins comprising one or more peptides with a high confidence score (95%) and a low false discovery rate (FDR) (estimated local FDR of 1%) were considered positively identified.

## Co-IP and western blotting

Co-IP assays were performed using transfected COS-7 cells or HEK293 cells, harvested in ice-cold PBS1X. Cell pellet was resuspended in either lysis buffer (NaCl 150 mM, Tris-HCl pH 7.2 50 mM, NP-40 1%, Desoxycholate Na 1%) or milder lysis buffer (NaCl 150 mM, Tris-HCl pH 7.2 50 mM, NP-40 0.5%, glycerol 10%) with 1X protease inhibitor cocktail (Roche) and incubated for 1 hr at 4°C under agitation. Cell lysates were cleared by centrifugation at 15 000 g for 20 min at 4°C. Supernatants were incubated with either GFP-TRAP (Chromotek) or HA-coupled beads (Sigma) as indicated for 1 or 2 hr at 4°C. The beads were collected and washed five times with 1 ml of ice-cold lysis buffer before SDS-PAGE. Eluates were loaded on 10 or 12% SDS-PAGE. After migration, proteins were transferred onto a P 0.45 µm PVDF or 0.45 µm nitrocellulose Amersham Hybond membranes (GE Healthcare) and immunoblotted with according antibodies. IP were repeated three times with anti-GFP for Cby-GFP/HA-Fam92 and once each (anti-HA or anti-GFP) for Cby-GFP/HA-Dzip1 and IP was performed once with anti-GFP for GFP-Dzip1/V5-Fam92/HA-Cby.

Primary antibodies used for western blotting were the following: rat anti-HA (1:5000; Roche), rabbit anti-GFP (1:10000; Abcam) and mouse anti-V5 (1:5000; Invitrogen). HRP-conjugated secondary antibodies were the following: goat anti-rabbit (1:10000; Biorad), goat anti-mouse (1:3000; Biorad) and goat anti-rat (1:20000; Sigma). Membranes were visualized using ECL prime from GE Healthcare.

## Fly stocks and maintenance

The *cby[1]* mutant and *cby::Tomato* transgene were previously described (*Enjolras et al., 2012*). The *cep290::GFP* transgene was a gift from T. Avidor-Reiss (University of Toledo, USA; *Blachon et al., 2009*; *Basiri et al., 2014*). CG6652::GFP, *mks1::GFP* and *unc::mKate* transgenes have been described in *Vieillard et al. (2016)*. *sas4[s2214]* mutant flies were kindly provided by R. Basto (*Basto et al., 2006*). Flies were raised on standard media between 21°C and 25°C. Df(3R)Exel8178 (*dzip1* deficiency), Df(2L)Exel6033 (*fam92* deficiency) were obtained from the Bloomington Stock Center and uncover the 95F8-–96A6 and 33E4-–33F2 cytological interval respectively. *cep290[0153-G4]* (Bloomington Drosophila Stock Center) harbors a piggy-bac insertion in the beginning of exon 11.

## Immunofluorescence and confocal microscopy

Testes from young adult flies or pupae were dissected in PBS1X, fixed 15 min in PBS1X/PFA 4% and either whole mounted or squashed between coverslip and slide and frozen in liquid nitrogen. Coverslip was removed and slides were soaked 2 min in ethanol 100% at −20°C. Testes were treated 15 min in PBS1X/Triton 0.1% (PBT) and blocked 2 hr in PBT/BSA 3%/NGS 5%.

Primary antibodies were incubated in blocking buffer overnight at 4°C. Samples were then washed in PBS1X and incubated 2 hr in secondary antibodies diluted in PBS1X. Slides were washed in PBS1X and rinsed in ultrapure water. Slides were mounted using Vectashield containing Hoescht 1:1000.

Antennae were processed as previously described (*Vieillard et al., 2015*). Briefly, *Drosophila* heads from 38 to 45 hr pupae were dissected in PBS1X, fixed 1 hr in PBS1X/PFA 4% and washed in PBS1X. Antennae were permeabilized 1 hr in PBS1X/Triton 0.3%, blocked 1 hr in PBS1X/Triton 0.3%/BSA 3%/NGS 5% and incubated in primary antibodies diluted in blocking solution for 48 hr at 4°C. Samples were washed three times in PBS1X and incubated in secondary antibodies diluted in PBS1X for 48 hr at 4°C. Antennae were washed three times in PBS1X and mounted in Vectashield.

For the whole mount staining of antennae, *Drosophila* heads from 38 to 45 hr pupae were dissected in PBS1X/0.3% Triton X-100 and fixed in in PBS1X/0.3% Triton X-100/4% PFA for 1 hr at RT. After a few rinses, samples were incubated 1 hr at RT in PBS1X/0.3% Triton X-100/BSA 0.1% and then in phalloidin FluoProbes 505 or 547 (Interchim) diluted 1:200 in PBS1X/0.3% Triton X-100/BSA 0.1% at 4°C for 48 hr. The samples were washed 3 times for 15 min in PBS1X and mounted in

Vectashield. All fluorescent observations were performed on at least five different pairs of testes or antennae. Quantifications were performed on at least three independent experiments.

Most slides were imaged using either the IX83 microscope from Olympus equipped with an iXon Ultra 888 EMCDD camera from Andor and the IQ3 software from Andor. The PlanApo N Apochromat 60 × 1.42 NA objective from Olympus was used for all acquisitions. Some slides were imaged using an SP5X confocal laser scanning microscope (Leica Biosystems) equipped with the Application Suite software (Leica Biosystems). An HCX Plan-Apochromat CS 63 × 1.4 NA objective (Leica Biosystems) was used for all acquisitions.

All images were processed with ImageJ. Figures were created with Adobe photoshop. Only contrasts and offset were adjusted.

## 3D-SIM microscopy

Testes or antennae were squashed on 12 diameter round coverslip with a 44 × 60 mm overlaying coverslip. Immunofluorescence protocols were the same as above using PFA. Images were acquired using the Elyra PS.1 system from Zeiss (Carl Zeiss, AG, Jena) equipped with a PCO edge 5.5 camera and the ZEN 2012 SP2 software (black edition). The objective used for all acquisitions is a Plan-apochromat 63 × 1.4 NA.

## Antibodies

The antibodies used were the following: mouse anti-Futsch/22C10 (1:250; DSHB = 22C10), mouse anti-γ-Tubulin (1:500; Sigma), rabbit anti-GFP (1:1000; Abcam), guinea pig and rat anti-Asterless (1:50000; gift from C. Rogers, University of Arizona, Tucson, USA; *Klebba et al., 2013*; *McLamarrah et al., 2018*).

The following secondary antibodies were used (all at 1:1000 dilution): goat anti-mouse Alexa 594, goat anti-rabbit Alexa 488 or Alexa 647, goat anti-guinea pig Alexa 488 or Alexa 594, goat anti-rat Alexa 488 or Alexa 555 or Alexa 647 (Invitrogen) and donkey anti-guinea pig Alexa 647 (Jackson Immuno Research).

## Quantifications

Bang assays were performed as previously described (*Enjolras et al., 2012*). Approximately 20 staged female flies were placed in graduated tubes (8 cm) and banged on the table at $t = 0$. The % of flies that reached each quarter of the tube was counted after 30 s. three different batchs were quantified for each genotype.

Quantifications of mother versus daughter centrioles were based on the principle that the daughter centriole, being nucleated from the mother centriole, is orthogonally positioned on the lateral wall of the mother centriole. Quantifications of TZ protein intensities was performed using ImageJ and by measuring the sum of pixel intensity in a defined region encompassing the centrioles. Background intensity was measured by measuring the sum of pixel intensity in an area close to and devoid of centrioles and then subtracted to TZ protein intensities.

## Statistics

Results of fluorescence intensity quantifications are represented as scatter plots with the mean and SD on all figures. Statistical significance was determined by two-tailed unpaired parametric Student's t test (*Figure 4A*, fertility assay; *Figure 5D*, Cby-Tom; *Figure 6A*, Cep290-GFP in *fam92*[1]) or nonparametric Mann-Withney's test when variances were not comparable (all other intensity quantifications of *Figure 5D* and *Figure 6A–C*). Results of phenotypic proportions are represented as contingency bar graph and statistical significance was determined by two-tailed Chi-square (*Figures 5D* and *6A*). (Prism six software; ns, $p > 0.05$; *, $p \leq 0.05$; **, $p \leq 0.01$; ***, $p \leq 0.001$; ****, $p \leq 0.0001$).

## Electron microscopy

Samples were processed as previously described (*Enjolras et al., 2012*; *Vieillard et al., 2015*). Observations were performed on at least two independent tissue samples.

## Acknowledgements

We thank T Avidor-Reiss for the *cep290::GFP Drosophila* stock and C Rogers for sharing the rat and guinea pig anti-Asterless. Many thanks to E Goillot for helpful discussions and M NGassa for technical assistance. This project was supported by a grant from the Fondation pour la Recherche Médicale, FRM DEQ20131029168 and the Agence Nationale de la Recherche (ANR Divercil). JAL was supported by the FRM DEQ20131029168. CA was supported by a PhD fellowship from the University Claude Bernard Lyon-1. We thank the PLATIM, CiQLE and CTμ platforms of the LyMIC for confocal and EM microscopy. We are grateful to Elodie Chatre for helpful discussions.

Mass spectrometry experiments were carried out using facilities of the Plaform 'Proteomic Imaging and molecular Interactions' (PP2I) of Montpellier.

## Additional information

### Funding

| Funder | Grant reference number | Author |
| --- | --- | --- |
| Agence Nationale de la Recherche | DIVERCIL | Jean-André Lapart<br>Elisabeth Cortier<br>Jean-Luc Duteyrat<br>Céline Augière<br>Julie Jerber<br>Joëlle Thomas<br>Bénédicte Durand |
| Fondation pour la Recherche Médicale | FRM DEQ20131029168 | Jean-André Lapart<br>Elisabeth Cortier<br>Jean-Luc Duteyrat<br>Céline Augière<br>Julie Jerber<br>Joëlle Thomas<br>Bénédicte Durand |
| Université Claude Bernard Lyon 1 | PhD Fellowship | Céline Augière |

The funders had no role in study design, data collection and interpretation, or the decision to submit the work for publication.

### Author contributions

Jean-André Lapart, Validation, Investigation, Visualization, Writing—original draft, Writing—review and editing; Marco Gottardo, Validation, Investigation, Visualization, Writing—review and editing; Elisabeth Cortier, Jean-Luc Duteyrat, Validation, Investigation, Visualization, Methodology; Céline Augière, Validation, Investigation; Alain Mangé, Formal analysis, Validation, Investigation, Writing—original draft; Julie Jerber, Validation, Investigation, Methodology; Jérôme Solassol, Resources, Validation, Project administration; Jay Gopalakrishnan, Resources, Validation, Writing—review and editing; Joëlle Thomas, Conceptualization, Supervision, Validation, Investigation, Writing—original draft, Writing—review and editing; Bénédicte Durand, Conceptualization, Supervision, Funding acquisition, Validation, Investigation, Writing—original draft, Project administration, Writing—review and editing

### Author ORCIDs

Jean-André Lapart https://orcid.org/0000-0001-9167-8391
Alain Mangé https://orcid.org/0000-0002-1566-9407
Joëlle Thomas https://orcid.org/0000-0002-0461-6131
Bénédicte Durand https://orcid.org/0000-0002-8530-0613

### Decision letter and Author response

Decision letter https://doi.org/10.7554/eLife.49307.sa1
Author response https://doi.org/10.7554/eLife.49307.sa2

## Additional files

### Supplementary files

• Supplementary file 1. Proteins identified by mass spectrometry from two rounds of LAP-Tag-CBY1 purification. Proteins in red were specifically retrieved in both replicates with LAP-CBY1 but not with LAP-GFP. Proteins in black were specifically retrieved in only one replicate with LAP-CBY1 but not with LAP-GFP. DZIP1L and FAM92A1 are highlighted in yellow

• Supplementary file 2. Primers used in this study.

• Transparent reporting form

### Data availability

All data generated or analysed during this study are included in the manuscript and supporting files.

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
