## [Decision Letter]

Thank you for submitting your article "Dzip1 and Fam92 form a ciliary transition zone module with cell type specific roles in *Drosophila*" for consideration by *eLife*. Your article has been reviewed by three peer reviewers, and the evaluation has been overseen by a Reviewing Editor and Anna Akhmanova as the Senior Editor. The reviewers have opted to remain anonymous.

The reviewers have discussed the reviews with one another and the Reviewing Editor has drafted this decision to help you prepare a revised submission.

I have left the original reviews below. But as you will note, the majority of the comments ask you to please revise the text so that the conclusions more accurately reflect the data and to avoid over-interpretation. Additional comments require re-analysis of the data and presentation of additional images. However, in the post-review discussion, all reviewers concur that as currently presented, there is insufficient experimental proof that Dzip1/Fam92/Cby are in a 'biochemical module', and ask that you perform additional experiments to support this statement.

Reviewer #1:

The ciliary transition zone (TZ) is crucial for ciliogenesis, and mutations in genes encoding TZ proteins lead to ciliopathies. Two pairs of recently discovered ciliary TZ proteins in mammalian cells are DZIP1/1L and FAM92A/B, both of which interact with the previously identified TZ protein CBY1. In previous studies, these proteins were shown to have roles in ciliogenesis and hedgehog signaling. However, their specific functions in these processes and their relationship to each other are poorly understood.

In this manuscript, the authors provide the first analysis of the *Drosophila* orthologs of these proteins, Dzip1 and Fam92. In both chordotonal (Ch) neurons and developing male germ cells, these proteins are needed to restrict localization of the TZ protein Cep290 to the TZ at the base of the cilium. However, Dzip1 and Fam92 otherwise play distinct roles in the two types of ciliated cells. In Ch neurons, they are needed for normal ciliogenesis, with *dzip1* mutants lacking cilia entirely and fam92 mutants having reduced numbers of cilia. Basal bodies (BB) are still associated with the tips of dendrites in these cells. However, Dzip1 and Fam92 are needed for docking of BB at the plasma membrane in primary spermatocytes. Interestingly, in *dzip1* and *fam92* mutants, docking of daughter centrioles is affected more severely than docking of mother centrioles, and the undocked centrioles form elongated cytoplasmic structures marked by the *Drosophila* lebercilin ortholog CG6652. Electron micrographs are consistent with the idea that Dzip1 and Fam92 are needed for formation of a cap that prevents this aberrant elongation. In addition, the authors demonstrate that 1) Cep290 acts upstream to ensure proper localization of Dzip1 and Fam92, 2) Dzip1 itself is needed for robust localization of Cby, Fam92 and the Meckel Syndrome protein Mks1, and 3) Cby and Fam92 are reciprocally required for each other's localization at the TZ.

Overall, the authors' results nicely lay out the cell type-specific requirements, localization and regulatory interactions of this group of TZ proteins. However, there are several caveats to the authors' claims that will need to be resolved before the manuscript is suitable for publication.

1) The authors claim "that Dzip1/Fam92/Cby form a biochemical module at the ciliary transition zone", and that they "all interact to form a biochemical complex". However, their biochemical data are limited to showing coIPs between Cby and Fam92 or Cby and Dzip1, but not between Fam92 and Dzip1. To support their conclusions, Fam92 and Dzip1 should be tested for their ability to coIP. Even if they do not coIP when co-expressed, they might form a tripartite complex with Cby, an idea that could be tested by co-expressing all three proteins and pulling on them sequentially, using different antibodies, to determine whether they interact indirectly as part of a multiprotein complex containing Cby.

2) It is unclear why the authors refer to CG6652-GFP as an axonemal marker (subsection “Dzip1 and Fam92 cap the basal body and facilitate its docking to the plasma membrane in spermatocytes”, first paragraph). In wild-type spermatocytes, CG6652-GFP is restricted to the distal tips of the BB, whereas in fam92, *dzip1* and *cep290* mutants, CG6652-GFP localizes to an extended structure that could either be an axoneme or an elongated TZ. I could not find evidence that CG6652-GFP localizes to axonemes in differentiating spermatids, either in the current manuscript or in the authors' previously published paper (Vieillard et al., 2016). A supplementary micrograph illustrating CG6652-GFP localization in whole-mount wild-type testes would be informative in this regard.

3) Given the authors' observation that BB are defective in docking in 8/13 Dzip1 and Fam92 mutant spermatocytes (subsection “Tissue specific functional properties of the TZ proteins in BB docking”, first paragraph) and their conclusion "that Dzip1 and Fam92 are at least involved in the initial steps of BB docking", it seems contradictory to claim that Dzip1 and Fam92 are involved in "maintaining basal body/membrane docking in *Drosophila* testes". The latter statement should be reworded to make clear that Dzip1 and Fam92 are needed for docking to initiate/occur (rather than be maintained).

4) Based on the data provided, it seems unclear whether "Dzip1 and Fam92 form a functional module which caps centrioles" or if their role in this process is indirect. It would be necessary to perform immuno-EM to determine if these proteins are physically associated with the centriolar cap. Alternatively, the authors should soften their conclusion regarding the function of Dzip1 and Fam92 in this process.

5) Supplementary file 1 and 2 are missing, so it is not possible evaluate the CBY1-associated proteins identified by mass spectrometry (subsection “Dzip1/Fam92/Cby form a complex module at the ciliary transition zone in *Drosophila*”, first paragraph).

Reviewer #2:

This is an interesting paper that requires some modification to remove several statements that over-interpret the data.

The authors show by that Cby, Dzip1 and Fam82 co-IP and from this claim the proteins form a biochemical complex. However, there is no evidence presented showing how these proteins might physically interact – if indeed they do physically interact. The authors go on to say that the three proteins perfectly co-localise – referring to Figure 1C. This reader is not able to see this perfect co-localisation. Thus, the conclusion reached at the end of this sections – that "Dzip1, Fam82 and Cby form a biochemical module" has not been demonstrated.

The difference in the strength of the mutant phenotypes for the *dzip1* and fam82 also seems to make the notion that their gene products form a biochemical entity appear unlikely. The authors report that a hypomorphic allele of *dzip1* has a strong uncoordinated phenotype whereas what appears to be a null allele of *fam92* exhibits a weaker phenotype. It is difficult to relate these phenotypes to any biochemical module. Reflecting the severity of the phenotype, *dzip1* mutants lack neurosensory cilia whereas fam82 flies could have with zero or one cilium in the scolopidia. This is a curious phenotype that perhaps suggests some mother – daughter difference in one of the cell generations leading to the mature scolopidium.

The authors then present convincing evidence that Dzip1 is needed to recruit both Cby and Fam92… and that Fam82 is required to recruit Cby but not Dzip1. They also see that in the absence of Dzip1, the domain of Cep290 staining is expanded. As this reader understands the text (clarification required), there "a few" (please define) examples of expanded Cep 290. Finally, in a strong Cep290 mutants both Dzip1 and Fam92 are reduced. This is interpreted to mean that Cep290 is required to recruit Dzip1 and Fam92 and these feed back to regulate localisation of Cep290.

Finally, the authors examine mutant defects in spermatocytes. My reading of this – although the text could be clarified – is that in both *fam92* and *dzip1* mutants, many centriole pairs show axonemal extensions from just one of the two centrioles. This is a particular interesting part of the phenotype that could be further characterised – or at least discussed in relation to the potential mother-daughter differences seen in the sensory cilia.

Reviewer #3:

The manuscript by Lapart et al. identifies Dzip1 and Fam92 as Chibby (Cby) interactors and convincingly show that Dzip1 and Fam92 localize and function at the transition zone and basal body is essential for ciliogenesis. In spermatocytes, authors make the exciting and novel discovery that *fam92* and *dzip1* mutants display an asymmetric phenotypic difference between mother and daughter centriole, with the daughter being more sensitive to TZ perturbation. This manuscript is supported by strong data, high quality imaging, and eloquently writing and will appeal to the readers of *eLife* who are interested in cilia, cell biology, germ cells, and human genetic diseases.

---

## [Author Response]

Reviewer #1:[…] Overall, the authors' results nicely lay out the cell type-specific requirements, localization and regulatory interactions of this group of TZ proteins. However, there are several caveats to the authors' claims that will need to be resolved before the manuscript is suitable for publication.1) The authors claim "that Dzip1/Fam92/Cby form a biochemical module at the ciliary transition zone", and that they "all interact to form a biochemical complex". However, their biochemical data are limited to showing coIPs between Cby and Fam92 or Cby and Dzip1, but not between Fam92 and Dzip1. To support their conclusions, Fam92 and Dzip1 should be tested for their ability to coIP. Even if they do not coIP when co-expressed, they might form a tripartite complex with Cby, an idea that could be tested by co-expressing all three proteins and pulling on them sequentially, using different antibodies, to determine whether they interact indirectly as part of a multiprotein complex containing Cby.

We agree with the reviewer that our first results did not demonstrate that all three proteins interact together in a same complex, but only that Cby interacts with either Fam92 or Dzip1. We have performed supplementary co-IPs and observed that Dzip1 and Fam92 do not apparently interact together, but when expressed with Cby, immunoprecipitation of Dzip1 pulls down both Cby and Fam92, suggesting that all three are present in one complex. These data are now included in Figure 1—figure supplement 1D and described in the first paragraph of the Results section.

Nevertheless, we agree that the term biochemical module might be too strong as our functional data also indicate that specific functions can be attributed to Dzip1 compared to Fam92 or Cby (see comments to reviewer 2 below). We have thus modified the text accordingly.

2) It is unclear why the authors refer to CG6652-GFP as an axonemal marker (subsection “Dzip1 and Fam92 cap the basal body and facilitate its docking to the plasma membrane in spermatocytes”, first paragraph). In wild-type spermatocytes, CG6652-GFP is restricted to the distal tips of the BB, whereas in fam92, dzip1 and cep290 mutants, CG6652-GFP localizes to an extended structure that could either be an axoneme or an elongated TZ. I could not find evidence that CG6652-GFP localizes to axonemes in differentiating spermatids, either in the current manuscript or in the authors' previously published paper (Vieillard et al., 2016). A supplementary micrograph illustrating CG6652-GFP localization in whole-mount wild-type testes would be informative in this regard.

Indeed, we never published the full description of CG6652-GFP distribution. We have included additional data on Figure 6—figure supplement 2 showing the expression of CG6652 in whole mount testes, illustrating the distribution of CG6652 along the sperm tail. In support of this axonemal association, CG6652-GFP is lost in *sas4* mutant elongated spermatids and we have added these observations on Figure 6—figure supplement 2 and in the first paragraph of the subsection “Dzip1 and Fam92 facilitate basal body docking to the plasma membrane in spermatocytes”.

3) Given the authors' observation that BB are defective in docking in 8/13 Dzip1 and Fam92 mutant spermatocytes (subsection “Tissue specific functional properties of the TZ proteins in BB docking”, first paragraph) and their conclusion "that Dzip1 and Fam92 are at least involved in the initial steps of BB docking", it seems contradictory to claim that Dzip1 and Fam92 are involved in "maintaining basal body/membrane docking in Drosophila testes". The latter statement should be reworded to make clear that Dzip1 and Fam92 are needed for docking to initiate/occur (rather than be maintained).

Indeed, the majority of basal bodies fail to initially dock and our writing was misleading. We have modified the statement as follows: “it also reveals tissue specific function of these proteins in priming basal body/membrane docking in *Drosophila* testes”.

4) Based on the data provided, it seems unclear whether "Dzip1 and Fam92 form a functional module which caps centrioles" or if their role in this process is indirect. It would be necessary to perform immuno-EM to determine if these proteins are physically associated with the centriolar cap. Alternatively, the authors should soften their conclusion regarding the function of Dzip1 and Fam92 in this process.

We agree with the reviewer that we do not prove that Dzip1 and Fam92 directly cap the centrioles. We have thus modified the text accordingly at several occurrences.

5) Supplementary file 1 and 2 are missing, so it is not possible evaluate the CBY1-associated proteins identified by mass spectrometry (subsection “Dzip1/Fam92/Cby form a complex module at the ciliary transition zone in Drosophila”, first paragraph).

We apologize for this problem and have provided the missing files.

Reviewer #2:This is an interesting paper that requires some modification to remove several statements that over-interpret the data.The authors show by that Cby, Dzip1 and Fam82 co-IP and from this claim the proteins form a biochemical complex. However, there is no evidence presented showing how these proteins might physically interact – if indeed they do physically interact. The authors go on to say that the three proteins perfectly co-localise – referring to Figure 1C. This reader is not able to see this perfect co-localisation. Thus, the conclusion reached at the end of this sections – that "Dzip1, Fam82 and Cby form a biochemical module" has not been demonstrated.

We agree with the reviewer that we did not demonstrate that all three physically interact together in a same complex but only that Cby and Fam92 or Dzip1 can be co-immuno-precipitated. We have performed supplementary co-IPs and observed that Dzip1 and Fam92 do not reside together in a complex stable enough to be immuno-precipitated. However, when co-expressed with Cby, immunoprecipitation of Dzip1 pulls down both Cby and Fam92, suggesting that all three are indeed present in one complex. These data are now included in Figure 1—figure supplement 1D and described in the first paragraph of the Results section.

We nevertheless modified the text to clearly state that we do not demonstrate physical interactions but only co-purification properties.

The difference in the strength of the mutant phenotypes for the dzip1 and fam82 also seems to make the notion that their gene products form a biochemical entity appear unlikely. The authors report that a hypomorphic allele of dzip1 has a strong uncoordinated phenotype whereas what appears to be a null allele of fam92 exhibits a weaker phenotype. It is difficult to relate these phenotypes to any biochemical module. Reflecting the severity of the phenotype, dzip1 mutants lack neurosensory cilia whereas fam82 flies could have with zero or one cilium in the scolopidia. This is a curious phenotype that perhaps suggests some mother – daughter difference in one of the cell generations leading to the mature scolopidium.

We agree with the reviewer that the difference in the severity of the phenotypes of *dzip1* and *fam92* mutant flies suggest that at least Dzip1 has additional functions than only recruiting Fam92 and Cby. We anticipate that Dzip1 interacts also with other components, yet to be identified, that are also required to organize the TZ. We agree that if Dzip1, Fam92 and Cby likely interact in a ternary complex based on CoIP experiments, their range of action is not limited to these interactions. This is now more clearly stated in the Discussion section.

We thank the reviewer for the suggested hypothesis of mother-daughter difference in cell generation leading to the scolopidium, but we think that testing this hypothesis goes beyond the scope of this manuscript.

The authors then present convincing evidence that Dzip1 is needed to recruit both Cby and Fam92… and that Fam82 is required to recruit Cby but not Dzip1. They also see that in the absence of Dzip1, the domain of Cep290 staining is expanded. As this reader understands the text (clarification required), there "a few" (please define) examples of expanded Cep 290. Finally, in a strong Cep290 mutants both Dzip1 and Fam92 are reduced. This is interpreted to mean that Cep290 is required to recruit Dzip1 and Fam92 and these feed back to regulate localisation of Cep290.

We have clarified the text as requested by the reviewer.

Finally, the authors examine mutant defects in spermatocytes. My reading of this – although the text could be clarified – is that in both fam92 and dzip1 mutants, many centriole pairs show axonemal extensions from just one of the two centrioles. This is a particular interesting part of the phenotype that could be further characterised – or at least discussed in relation to the potential mother-daughter differences seen in the sensory cilia.

Yes indeed, we observed extensions more often on the daughter centriole as presented in the Results section and discussed. We added one sentence at the end of the Introduction and modified the sentence in the Results section to better emphasize on this result.